# Unexpectedly minor nitrous oxide emissions from fluvial networks draining permafrost catchments of the East Qinghai-Tibet Plateau

Liwei Zhang [1,7,8], Sibo Zhang[2,8], Xinghui Xia [1✉], Tom J. Battin [3], Shaoda Liu[1], Qingrui Wang [1], Ran Liu[4], Zhifeng Yang[2], Jinren Ni [5] & Emily H. Stanley [6]

Streams and rivers emit substantial amounts of nitrous oxide ($N_2O$) and are therefore an essential component of global nitrogen (N) cycle. Permafrost soils store a large reservoir of dormant N that, upon thawing, can enter fluvial networks and partly degrade to $N_2O$, yet the role of waterborne release of $N_2O$ in permafrost regions is unclear. Here we report $N_2O$ concentrations and fluxes during different seasons between 2016 and 2018 in four watersheds on the East Qinghai-Tibet Plateau. Thawing permafrost soils are known to emit $N_2O$ at a high rate, but permafrost rivers draining the East Qinghai-Tibet Plateau behave as unexpectedly minor sources of atmospheric $N_2O$. Such low $N_2O$ fluxes are associated with low riverine dissolved inorganic N (DIN) after terrestrial plant uptake, unfavorable conditions for $N_2O$ generation via denitrification, and low $N_2O$ yield due to a small ratio of nitrite reductase: nitrous oxide reductase in these rivers. We estimate fluvial $N_2O$ emissions of $0.432 - 0.463$ Gg $N_2O$-N yr$^{-1}$ from permafrost landscapes on the entire Qinghai-Tibet Plateau, which is marginal (~0.15%) given their areal contribution to global streams and rivers (0.7%). However, we suggest that these permafrost-affected rivers can shift from minor sources to strong emitters in the warmer future, likely giving rise to the permafrost non-carbon feedback that intensifies warming.

[1] Key Laboratory of Water and Sediment Sciences of Ministry of Education & State Key Laboratory of Water Environment Simulation, School of Environment, Beijing Normal University, Beijing, China. [2] Guangdong Provincial Key Laboratory of Water Quality Improvement and Ecological Restoration for Watersheds, School of Ecology, Environment and Resources, Guangdong University of Technology, Guangzhou, China. [3] Stream Biofilm and Ecosystem Research Laboratory, School of Architecture, Civil and Environmental Engineering, École Polytechnique Fédérale de Lausanne, Lausanne, Switzerland. [4] Department of Mathematics, Beijing Jiaotong University, Beijing, China. [5] Key Laboratory of Water and Sediment Sciences of Ministry of Education, College of Environmental Sciences and Engineering, Peking University, Beijing, China. [6] Center for Limnology, University of Wisconsin-Madison, Madison, WI, USA. [7] Present address: Sino-French Institute for Earth System Science, College of Urban and Environmental Sciences, Peking University, Beijing, China. [8] These authors contributed equally: Liwei Zhang, Sibo Zhang. ✉email: xiaxh@bnu.edu.cn

itrous oxide ($N_2O$) is a major stratospheric ozone destroyer and the third most important long-lived greenhouse gas (GHG)[1]. Sources of this powerful GHG are poorly constrained in general, and for global streams and rivers in particular[2]. Studies of lotic $N_2O$ dynamics have focused almost exclusively on human-impacted lowland systems across diverse climate zones, where anthropogenic nitrogen (N) enrichment has been linked to elevated $N_2O$ fluxes[3]. Unfortunately, data are extremely sparse for more pristine regions, including the rapidly changing permafrost-rich cryospheres at high altitudes and latitudes. As a result, the current estimate of global fluvial emissions of $291.3 \pm 58.6$ Gg $N_2O$-N yr$^{-1}$ has substantial uncertainty[4].

Massive amounts of organic carbon (OC, ~1014 Pg C) are stored in the top 3 m of Northern Hemisphere permafrost soils[5]. As these deposits thaw, ice-locked carbon is liberated and can be transported to adjacent running waters, where it becomes available for processing and loss to the atmosphere as carbon dioxide ($CO_2$) and methane ($CH_4$)[6–9]. Consequently, fluvial emissions of these gases in some cryosphere regions are now occurring at elevated and increasing rates[10,11]. However, the magnitude of $N_2O$ emissions from streams and rivers in regions where permafrost thaw is afoot is unknown despite the fact that Northern Hemisphere permafrost soils contain 67 Pg N to a depth of 3 m (excluding N pools in the active layer), and are evident or even substantial sources of $N_2O$[12]. Studies of Alaskan streams have emphasized sustained delivery of N from thawing permafrost soils resulting in elevated inorganic N[13,14] and often-supersaturated $N_2O$ concentrations in these receiving waters[13]. However, in the absence of direct measurements of $N_2O$ emissions from permafrost-affected streams and rivers, it is not clear if these results are representative and translate to overall elevated $N_2O$ emissions.

To address this knowledge gap, we provide a cross-regional and seasonal direct measurement of fluvial $N_2O$ concentrations and fluxes, generated in four headwater catchments that vary in altitude from 1650 to 4600 m and cover an area of ca. $73.6 \times 10^4$ km$^2$ on the East Qinghai-Tibet Plateau (EQTP; Fig. 1; Supplementary Fig. 1, Table 1 and 2), and identify likely mechanisms shaping $N_2O$ dynamics in these rivers. Areal $CH_4$ emissions from streams and rivers in this region are strongly affected by permafrost thaw, and are among the highest reported rates globally[11], suggesting that a similar effect may be underway for $N_2O$ in these rivers as well. This region is the largest cryosphere outside the Arctic and Antarctic[15], with vast Pleistocene-aged permafrost[16] that contains an estimated 1.8 Pg N in the upper 3 m of soil[17]. As the 'Water Tower of Asia', meltwater from the Qinghai-Tibet Plateau (QTP) feeds ten great Asian rivers and ubiquitous ponds, lakes, and wetlands[18]. Such alpine streams and rivers are tightly connected to permafrost soils[11] and have high turbulence-induced gas exchange[19], and these features potentially promote their capacity to process terrestrial N and sustain high $N_2O$ emissions. Further, many parts of the QTP have experienced significant human population growth over the last six decades accompanied by similar rapid increases in livestock[20], again adding to the potential for enhanced riverine $N_2O$ emissions. However, as we show below, these alpine permafrost-affected streams and rivers of the EQTP are unexpectedly minor atmospheric $N_2O$ sources. Yet there are signs that developing climate warming and enhanced anthropogenic influences may lead to a substantial increase in fluvial $N_2O$ emissions from high altitudes and latitudes in the coming decades. This study offers a guide to include permafrost streams and rivers in current and future $N_2O$ inventories, given their potentially important implications for $N_2O$ budget.

## Results and discussion

**Variability of $N_2O$ concentrations and fluxes**. All sampled streams and rivers were supersaturated on all dates (117.9–242.5%, $n = 342$ samples from 114 site visits) in $N_2O$ with respect to the atmosphere. Dissolved $N_2O$ concentrations fluctuated between 10.2 and 18.9 nmol L$^{-1}$ with an average of $12.4 \pm 1.7$ nmol L$^{-1}$, which is one-third of the global average[3]

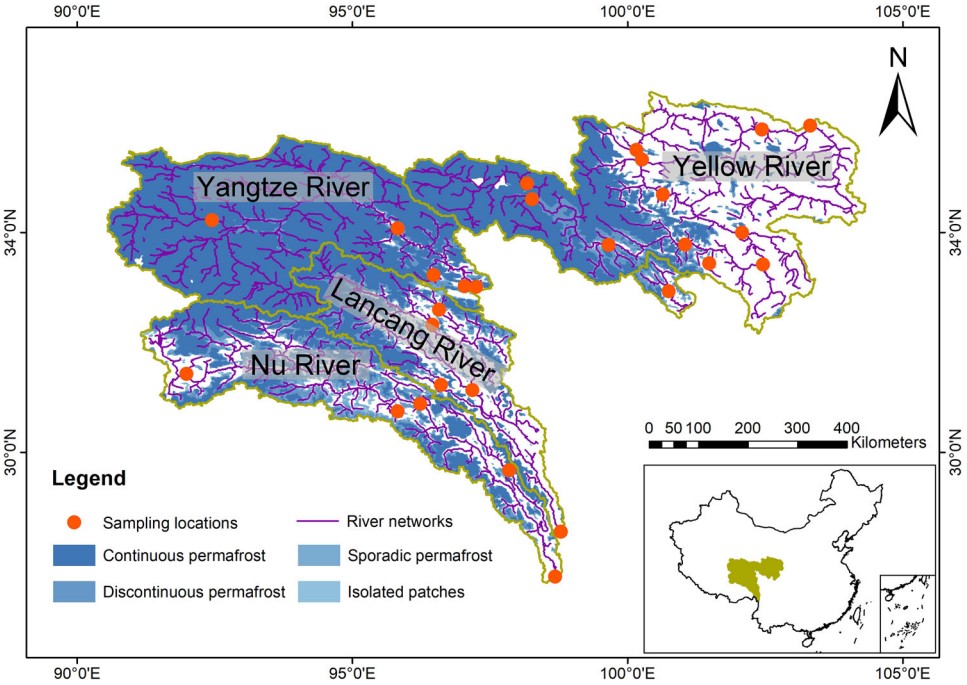

**Fig. 1 Map of the study sites in four headwater catchments on the East Qinghai-Tibet Plateau (EQTP), China.** Sampling sites within the four catchments. The blue shading represents permafrost extent on the EQTP (Data for the permafrost extent courtesy of ref. [57]). The inset highlights the study area on a map of China.

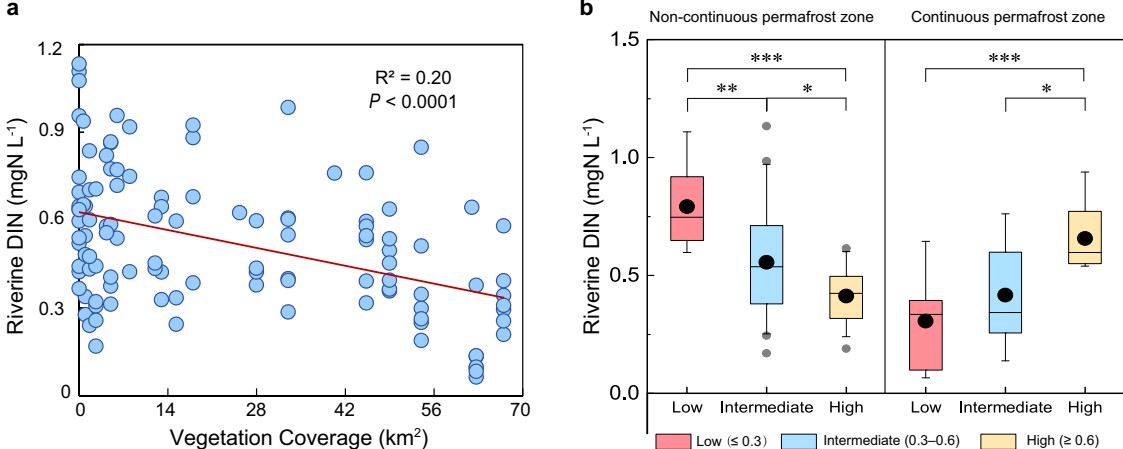

**Fig. 2 Effect of vegetation on riverine dissolved inorganic nitrogen (DIN). a** Correlation between vegetation coverage (see Methods) and riverine DIN for all sites across all permafrost categories. The red line represents the fit of a linear regression through the observed data. **b** Riverine DIN for each specific normalized difference vegetation index (NDVI) interval (≤0.3; 0.3–0.6; ≥0.6) in continuous and non-continuous permafrost zones across different seasons (one-way ANOVA with Tukey's post-hoc test: *$P < 0.05$; **$P < 0.01$; ***$P < 0.001$). Boxes represent the 25th and 75th percentiles, and error bars show the 95th percentiles. Black circles and horizontal lines indicate the arithmetic means and medians, respectively. Gray circles are outliers.

(37.5 nmol L$^{-1}$; Supplementary Table 3). Significantly higher N$_2$O concentrations were observed in spring ($P < 0.001$), followed by fall and summer (Supplementary Fig. 2a). Despite differences in catchment attributes including permafrost fraction and population densities (Supplementary Table 2), N$_2$O concentrations were not significantly different among the four river systems (Supplementary Fig. 2a).

Diffusive N$_2$O fluxes from EQTP rivers were predominantly positive (to the atmosphere), ranging from −14.0 to 40.6 µmol m$^{-2}$ d$^{-1}$ with an average of $9.4 \pm 6.2$ µmol m$^{-2}$ d$^{-1}$ ($n = 436$ samples from 114 site visits). This mean flux is an order of magnitude lower than the global average[3] (94.3 µmol m$^{-2}$ d$^{-1}$; Supplementary Table 3). Diffusive N$_2$O fluxes were similar in summer and fall, and significantly higher than those in spring ($P < 0.05$; Supplementary Fig. 2b). The asynchronous seasonal patterns between concentrations and fluxes are likely caused by water temperature and precipitation (Supplementary Discussion 1). As with concentration, no significant differences were found for N$_2$O diffusion among the four rivers (Supplementary Fig. 2b).

N$_2$O ebullition has seldom been documented in lotic ecosystems, although it can coincide with CH$_4$ bubble release[21]. Because of the presence of large organic reserves in surrounding permafrost, shallow water depths, and their exposure to low barometric pressure, streams and rivers on the EQTP are notable hotspots of CH$_4$ ebullition[11]. Thus, we had hypothesized that N$_2$O would also be entrained during the widespread processes of CH$_4$ bubble release. The mean N$_2$O ebullition rate from EQTP rivers was $0.74 \pm 2.47$ µmol m$^{-2}$ d$^{-1}$, and accounted for $4.1 \pm 11.9\%$ of total N$_2$O fluxes (diffusion + ebullition) across all sites. Despite this potential for high ebullition rates, accounting for this flux had a little overall effect, as the average total N$_2$O flux ($10.2 \pm 7.1$ µmol m$^{-2}$ d$^{-1}$) was still nine-fold lower than the global mean diffusive N$_2$O.

**Terrestrial processes modulating N$_2$O dynamics.** The availability of inorganic N is often the primary determinant of rates of N$_2$O production and emission in both permafrost-affected soils[12] and fluvial networks[2]. In the QTP, dissolved N released from both thawing permafrost soils and animal manure (Supplementary Discussion 2) is biologically available for plant uptake[22–24] or instead may be exported to river channels. Because plant growth is N-limited in this region[24], we predicted that an increase in

vegetation cover would result in greater plant uptake of terrestrial N, and consequently lower inputs to, and concentrations of N in streams and rivers, up to a point. Indeed, riverine dissolved inorganic N (DIN) concentrations decreased with increasing vegetation cover for all sites (Fig. 2a). However, mechanisms causing the observed decline in riverine DIN cannot be elucidated from vegetation coverage alone, as productivity and greenness per unit plant cover decline at higher elevations[25]. Thus, we also examined normalized difference vegetation index (NDVI, a measure of productivity and greenness) to indicate plant N uptake. Sites with high NDVI had low riverine DIN in non-continuous (namely discontinuous, sporadic, and isolated) permafrost zones (Fig. 2b), in line with the hypothesis of greater plant influence on riverine N availability when and where vegetation productivity and greenness were higher. In contrast, in areas with continuous permafrost, sites with high NDVI had high riverine DIN (Fig. 2b), suggesting that terrestrial N was sufficient to support plant growth and associated productivity/greening regardless of season, and indeed likely exceeded the N demand of plant community[26]. If so, then the surplus DIN can be exported to river corridors. Terrestrial N$_2$O can also be transported along with DIN to surrounding watercourses, and higher N$_2$O concentrations in some sites draining permafrost areas could be supported by transport of terrestrial N$_2$O and/or greater in situ N$_2$O production supported by more DIN inputs. Even so, EQTP streams and rivers collectively received reduced terrestrial N after plant uptake. DIN concentrations ($0.54 \pm 0.30$ mgN L$^{-1}$) in EQTP waterways were at the lower end of the range reported for global streams and rivers[3] (0.002–21.2 mgN L$^{-1}$) and constrained within a relatively narrow range. The low N$_2$O concentrations and fluxes are consistent with the low N availability in these rivers.

**Biogeochemical processes regulating N$_2$O dynamics.** Riverine N$_2$O concentrations could not be effectively predicted by simple linear regressions with environmental variables [$R^2 \leq 0.1$ in most cases, including dissolved oxygen (DO, $P > 0.05$, $R^2 = 0.004$) and NH$_4^+$ concentrations ($P < 0.001$, $R^2 = 0.1$); Supplementary Table 4]. However, we found a strong positive relationship between NO$_3^-$ and N$_2$O concentrations when DO saturation (% O$_2$) was undersaturated (<100%) in the water column (Fig. 3a). This result was validated by a regression tree analysis that

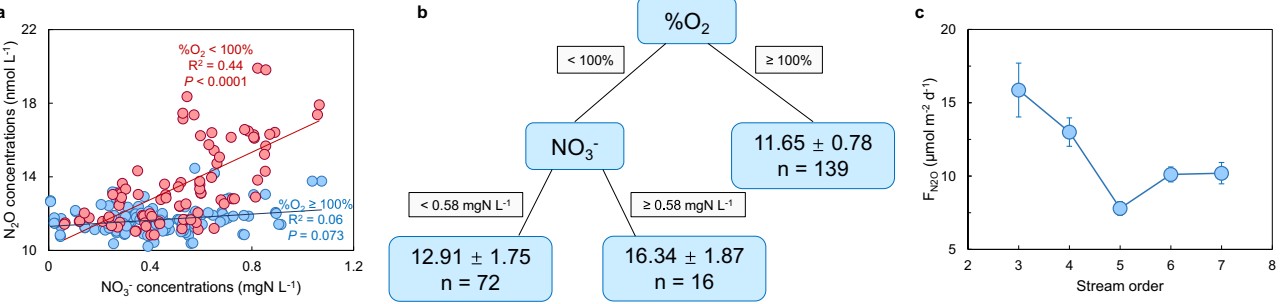

**Fig. 3 N₂O as functions of environmental variables. a** $N_2O$ concentrations as functions of $NO_3^-$ concentrations for samples with supersaturated $O_2$ (blue symbols) and undersaturated $O_2$ (red symbols). **b** Regression tree describing predictors of $N_2O$ concentrations in EQTP rivers. Parameters entering the model were $\%O_2$ and $NO_3^-$. Values at the ends of each terminal node indicate the $N_2O$ concentrations (nmol $L^{-1} \pm 1\,SD$) and number of observations (n). Cross-validated relative error was $1.70 \pm 0.02$ and $R^2$ was 0.56. **c** $F_{N_2O}$ in relation to stream order across EQTP rivers. All error bars represent $\pm 1$ SE.

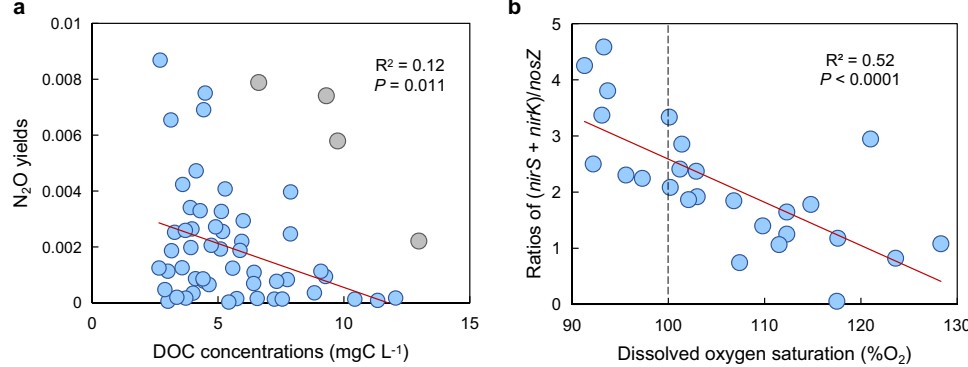

**Fig. 4 Effect of environmental variables on N₂O yields and ratios of nitrite reductase: nitrous oxide reductase [(*nirS* + *nirK*)/*nosZ*]. a** Correlation between dissolved organic carbon (DOC) concentrations and $N_2O$ yields. Grey points denote samples affected by reservoirs, and were excluded from the correlation analysis. **b** Correlation between dissolved oxygen saturation $(\%O_2)$ and ratios of $(nirS + nirK)/nosZ$. The red lines represent the fit of a linear regression through the observed data and the vertical dashed line denotes the boundary between super- and sub-saturated dissolved oxygen.

identified $\%O_2$ as the primary control on $N_2O$ concentration, and that higher $N_2O$ concentrations occurred when $\%O_2 < 100\%$ and $NO_3^- \geq 0.58\,\text{mgN L}^{-1}$ (Fig. 3b). The tree had a greater explanatory power ($R^2 = 0.56$) than the simple linear regression of $NO_3^-$ and $N_2O$ concentrations ($R^2 = 0.23$; Supplementary Table 4). $N_2O$ concentrations were uniformly low and weakly related to $NO_3^-$ when $\%O_2 \geq 100\%$ (Fig. 3a), suggesting that $N_2O$ present at these sites was derived from rare surface sediment patches in the channel that maintain hypoxic-anoxic conditions despite abundant $O_2$ in the water column or from external sources including inputs of dissolved $N_2O$ from permafrost soils, intermediate runoff (Supplementary Fig. 3), and upwelling groundwater[27] to maintain such low to modest but super-saturated $N_2O$ concentrations. Regardless of the specific source, this result means that the widespread occurrence of well-oxygenated overlying waters of EQTP rivers (139 of 227 samples) limits the extent of hypoxic-anoxic regimes needed for $N_2O$ generation via denitrification.

The emission factor $EF_{5\text{-}r}$ ($EF_{5\text{-}r} = N_2O\text{-}N/NO_3^-\text{-}N$) is a surrogate for the conversion of riverine $NO_3^-$ to dissolved $N_2O$[28]. Mean $EF_{5\text{-}r}$ for EQTP rivers (0.17%) was lower than both the global average (0.22%; Supplementary Table 3) and Intergovernmental Panel on Climate Change (IPCC) default value (0.26%) and is indicative of a small portion of riverine $NO_3^-$ being converted to dissolved $N_2O$ in EQTP rivers. According to our regression tree analysis, the discrepancies between $EF_{5\text{-}r}$ for EQTP rivers and IPCC estimate corroborate past studies that a simple linear $NO_3^-$ model does not adequately predict actual $N_2O$[29,30], because $N_2O$ concentrations will not necessarily

increase with $NO_3^-$ loads in oxic environments. We therefore recommend that the IPCC methodology should be revised to consider nonlinear relationships or interactions among multiple environmental variables.

**Microbial processes underlying N₂O dynamics.** $N_2O$ yield $[\Delta N_2O/(\Delta N_2O + \Delta N_2) \times 100\%]$ is a useful metric of relative $N_2O$ generation[27], and in EQTP rivers, the $N_2O$ yield (0.003–0.87%, average 0.23%) was 11 times lower than has been reported for lotic settings (0.01–53.8%, average 2.47%; Supplementary Table 5). This small percentage indicates that N processing in EQTP streams and rivers predominantly generates dinitrogen ($N_2$) instead of $N_2O$. Laboratory determination of benthic $N_2O$ production rates confirmed the consistent conversion of $N_2O$ to $N_2$, as these rates were negative ($N_2O$ was consumed) for two-thirds of the sites (Supplementary Table 6). Low $N_2O$ yields have been associated with the availability of ample OC to support the complete reduction of $NO_3^-$ to $N_2$[2], and in addition to supplying N, thawing permafrost is also a source of biolabile OC to EQTP streams and rivers[11] (Fig. 4a).

Examination of relative gene abundances involved in $N_2O$ production and consumption provides further insights regarding the reason for the low $N_2O$ yield in EQTP rivers. The key enzymes for $N_2O$ production are two types of nitrite reductase (*nirS* and *nirK*)[31], and $N_2O$ consumption is mediated by clade I and II nitrous oxide reductase (*nosZ*) which catalyzes $N_2O$ reduction to $N_2$[32]. A high ratio of $(nirS + nirK)/nosZ$ indicates an amplified capacity for $N_2O$ production relative to its loss, leading

to high $N_2O$ concentrations in the water column. This ratio for EQTP riverbed sediments (average 1.96) was far below values reported from other lotic settings worldwide that vary from 2.16 to $3.24 \times 10^6$ (average 19.8; Supplementary Table 7), providing compelling evidence for a molecular basis for the low $N_2O$ yield in EQTP rivers. Unexpectedly, we also found a negative correlation between the $(nirS + nirK)/nosZ$ ratio and %$O_2$ (Fig. 4b; Supplementary Discussion 3), illustrating that the microbial community increasingly favors the reduction of $N_2O$ to $N_2$ as DO saturation increased[33].

**Physicochemical processes governing $N_2O$ dynamics**. To understand potential controls on $N_2O$ fluxes ($F_{N_2O}$), we used stepwise regression to assess the relationships between $F_{N_2O}$ and multiple environmental variables known to influence $F_{N_2O}$. The analysis showed that %$O_2$, pH, water temperature, total phosphorus, and $NO_3^-$ all had significant but weak relationships with $F_{N_2O}$ ($P < 0.001$, $R^2 = 0.14$; Supplementary Table 8).

Local hydrogeomorphology is, at least qualitatively, a reliable predictor of $F_{N_2O}$ downstream trends[2]. Along the longitudinal continuum, $F_{N_2O}$ was highest in $3^{rd}$-order (headwater) streams, declined in $4^{th}$- and $5^{th}$-order (medium-sized) rivers, and was slightly elevated in $6^{th}$- and $7^{th}$-order (large) rivers (Fig. 3c). The decline in flux from $3^{rd}$- to $5^{th}$-order streams may reflect the reduced perimeter-to-surface-area ratio (ratio of wetted perimeter to cross-sectional area) and hyporheic exchange rates (exchange rates of dissolved substances between surface water and groundwater beneath and alongside the river channel) with increasing stream order[34–36], while the increase in $6^{th}$- and $7^{th}$-order sites might be due to increasing riverine DIN concentrations[2] (Supplementary Fig. 4). Furthermore, increased suspended sediment loads can enhance $N_2O$ generation in larger turbid channels, as suspended particles provide micro-niches that support N transformations[37], and thus facilitate $N_2O$ production in the water column[2,36]. Suspended sediment concentration increased with stream order for these rivers[11], lending support to the hypothesis that this mechanism contributes to higher fluxes observed in $6^{th}$- and $7^{th}$-order channels.

**Regional and global implications**. Based on our flux measurements, we estimated that EQTP $3^{rd}$- to $7^{th}$-order streams and rivers emitted 0.206 Gg $N_2O$-N $yr^{-1}$ ($5–95^{th}$ percentiles: 0.129–0.291 Gg $N_2O$-N $yr^{-1}$, 2603 $km^2$ of river channel area). Our upscaling did not include $1^{st}$- and $2^{nd}$-order streams, which can contribute disproportionately high areal fluxes to overall fluvial $N_2O$ emissions[36,38]. Low-order streams are always well connected to continuous permafrost and hence should receive high N inputs while having reduced $N_2O$ solubility owing to high altitudes, and these conditions are expected to lead to high $N_2O$ fluxes. Based on this logic, we estimated a total emission of 0.275 Gg $N_2O$-N $yr^{-1}$ from $1^{st}$- to $7^{th}$-order streams and rivers ($5–95^{th}$ percentiles: 0.162–0.400 Gg $N_2O$-N $yr^{-1}$, 3049 $km^2$) by extrapolating the relationship in Fig. 3c to include $1^{st}$- and $2^{nd}$-order streams.

Despite large uncertainties due to a lack of observational data from $1^{st}$- and $2^{nd}$-order streams (Supplementary Discussion 4), the upscaling exercise enables us to place our estimates in a broader context of both regional $N_2O$ budgets and fluvial emissions at the global scale. The percentage of $N_2O$ in total GHG ($CO_2 + CH_4 + N_2O$) emissions expressed as $CO_2$ equivalents corresponded to 1.0% for EQTP $3^{rd}$- to $7^{th}$-order drainages, then dwindled to 0.4% for EQTP $1^{st}$- to $7^{th}$-order streams and rivers, falling within the range of pristine rivers (0.2–1.2%)[39,40]. These values contrast those from human-impacted fluvial

networks, where $N_2O$ percentages are generally much higher (2.8–13.9%, average 6.8%; Supplementary Table 9) due to often elevated inputs of fertilizer- or sewage-derived N that boost $N_2O$ emissions. Expressed as per unit stream/river and basin area, EQTP $1^{st}$- to $7^{th}$-order streams and rivers released a total of 0.08 t $N_2O$-N $km^{-2}$ $yr^{-1}$ and 0.32 kg $N_2O$-N $km^{-2}$ $yr^{-1}$, respectively, to the atmosphere, which are one order of magnitude lower than those from lotic systems worldwide [0.65 (range: 0.08–2.55) t $N_2O$-N $km^{-2}$ $yr^{-1}$ and 2.44 (range: 0.55–5.78) kg $N_2O$-N $km^{-2}$ $yr^{-1}$, respectively; Supplementary Table 9]. Applying the emission rate per unit stream/river and basin area to the entire QTP $1^{st}$- to $7^{th}$-order drainage networks, we obtained a riverine $N_2O$ emission of 0.432–0.463 Gg $N_2O$-N $yr^{-1}$, which is minor (~0.15%) given their areal contribution (0.7%) to global streams and rivers[41]. In addition, these $N_2O$ estimates are probably overestimated. The emission of $N_2O$ accumulated underneath the ice during winter was estimated to be 15% of the annual emission[39]. This ice-melt outgassing of winter $N_2O$ was included in the above annual $N_2O$ emissions; however, this flux may be very limited in permafrost-affected systems due to minimum N inputs from frozen soils in winter[42]. These alpine permafrost waterways emit large amounts of $CH_4$[11], but fortunately they are currently small contributors of $N_2O$ delivery to the atmosphere, demonstrating $CH_4$ and $N_2O$ dynamics are uncoupled within these systems.

Although QTP fluxes were small, existing global estimates do not effectively capture this natural fluvial source of $N_2O$, nor are the major drivers of $N_2O$ dynamics well known for these systems. Our study is a step forward in quantifying fluvial $N_2O$ evasion from a cryospheric biome, and highlights the unique dynamic nature of $N_2O$ concentrations and fluxes in high-altitude environments. Our finding that oxygen saturation was the first and primary correlate of $N_2O$ concentrations may also have broader implications for aquatic $N_2O$ dynamics in other high-altitudinal streams.

**Future fluvial $N_2O$ emissions under warming climate**. Temperatures are rising faster in high altitudes and latitudes than in other regions[43]. As warming continues, permafrost thaw is expected to increase, liberating substantial amounts of dissolved N[12]. As this process progresses into deeper soil layers below the rhizosphere, diminished plant uptake[24] should favor greater export of N to streamflow via deep flow paths[14,44,45]. Meanwhile, warmer water temperatures reduce gas solubility and enhance hypoxia and denitrification at the expense of anammox[46], directing more N towards denitrification, and concomitant $N_2O$ production and evasion to the atmosphere (Fig. 5). Moreover, the duration of the ice-free season across the cryosphere is rapidly increasing and will continue to increase[47], suggesting a potential proxy for riverine $N_2O$ release. Furthermore, human perturbations may bring an extra N burden to the cryosphere and exacerbate these impacts. Taken together, these processes might render streams and rivers draining permafrost catchments across the globe to become hotspots of $N_2O$ to the atmosphere in the future, leading to positive non-carbon climate feedback of currently unanticipated magnitude because of an increase in fluvial $N_2O$ production following the development of climate change and escalation of anthropogenic influence.

The high degree of spatiotemporal variability in riverine $N_2O$ observed here is likely to exist in other unexplored cryospheres. Further progress in understanding how aquatic ecosystems in these climate-sensitive regions will respond to ongoing global warming would benefit greatly from more $N_2O$ measurements from glacial and permafrost-affected lotic and lentic systems at high spatial and temporal resolution. Alongside $N_2O$ measurements, capturing the fate of thawed N in cryospheric aquatic

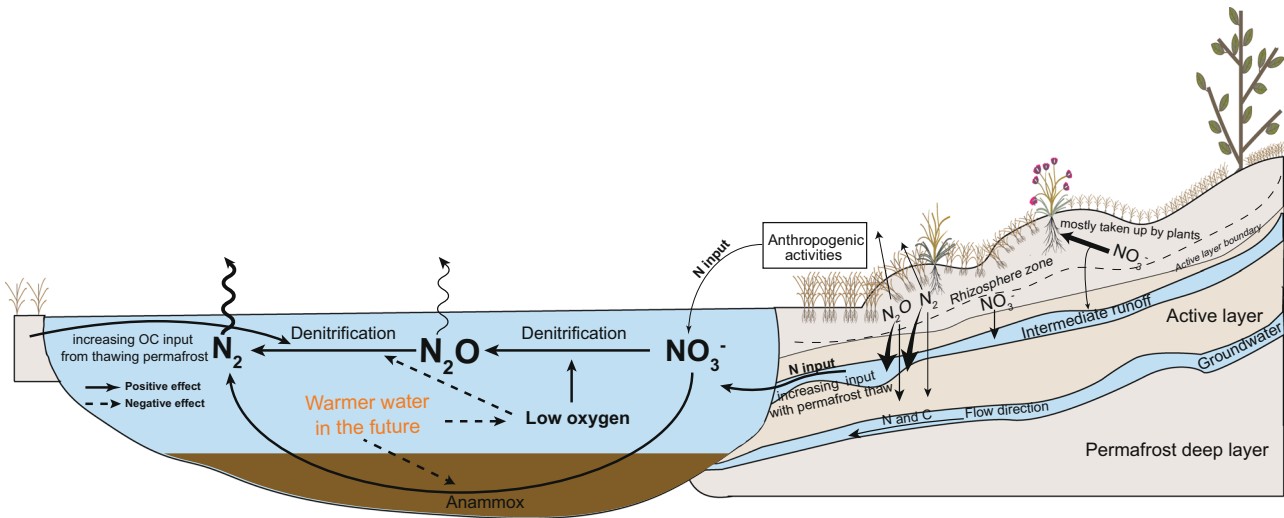

**Fig. 5 Conceptual model of potential fluvial N and/or $N_2O$ processes and pathways with deeper thaw in a warmer climate.** Future warming and associated permafrost thaw, together with enhanced human perturbations will increase terrestrial N input into surrounding river networks. Warming also raises water temperatures (highlighted in orange), promoting $N_2O$ production via denitrification with suppressing the reduction of $N_2O$ to $N_2$, eventually resulting in elevated $N_2O$ emissions.

systems are indispensable to stitching together pathways and processes into a holistic framework. It is time to improve our understanding of cryospheric aquatic $N_2O$ emissions in shaping the global $N_2O$ budget, and how this contribution might be altered in progressively warming high altitudes and latitudes.

## Methods

**Site description**. The ~73.6 × 10⁴ km² study area of the East Qinghai-Tibet Plateau (EQTP) is located between the boundary of the Loess Plateau (Gansu Province, 36° N) and the southern frigid limit (Yunnan Province, 28° N). This region is characterized by a high-relief landscape ranging from 1650 to 7000 m above sea level. The elevational gradient, along with abundant but variably distributed permafrost creates geomorphological, hydrological, vegetative, and climatological heterogeneities in the alpine landscape. The EQTP includes four great Asian rivers—the Yangtze, Yellow, Lancang-Mekong and Nu-Salween Rivers. We sampled a broad range of streams and rivers in these basins that ranged from Strahler order 3 to 7. All sampling sites were visited during the daytime in spring (May–June), summer (July–August), and fall (September–October) between 2016 and 2018. The Yellow River was sampled seven times, the Yangtze River four times, and the Lancang and Nu Rivers were both sampled on three dates. We grouped the sampling sites into four categories that represent different permafrost zones: continuous, discontinuous, sporadic, and isolated (Fig. 1). We then merged discontinuous, sporadic, and isolated permafrost zones together under the non-continuous permafrost group in response to different NDVI trends.

**$N_2O$ concentrations and fluxes**. Triplicate samples for dissolved $N_2O$ concentrations were collected by completely filling 120 mL glass serum bottles at wrist depth below the water surface at each site. After preserving samples with 0.5 mL saturated $ZnCl_2$ solution, the serum bottles were sealed with butyl stoppers, crimped with aluminum caps, and stored at ambient temperature in the dark. Local ambient air samples were also taken and used to back-calculate $N_2O$ concentration in water in equilibrium with the atmosphere. $N_2O$ concentrations were determined via the headspace equilibration method[48] on a gas chromatography equipped with an electron capture detection for $N_2O$ (Agilent 7890B GC-μECD). The partial pressure of $CO_2$ ($pCO_2$) in surface water was determined following our earlier work[11]. Sampling and analysis of dissolved $N_2$ concentrations are described in Supplementary Methods 1.

$F_{N_2O}$ was measured simultaneously with dissolved $N_2O$ concentration collection. Four floating chambers were held in place at each transect, covering depth gradients from the river bank to the mid-channel to capture the spatial heterogeneity. Measurements lasted for 1 h at each site, and 50 mL gas extracted from inside the chambers at 0, 5, 10, 20, 40, and 60 min intervals were injected into air-tight gas sampling bags for analysis in the laboratory by GC-μECD. These chambers were of the same size and shape and streamlined with a flexible plastic foil collar to minimize the effects of chamber-induced turbulence when measuring fluxes[49] and were covered with aluminum foil to reflect the sunlight and minimize internal heating.

Surface water and sediment samples were collected simultaneously with gas samples for physicochemical and microbial analyses at each site, respectively. Air temperature, air pressure, and wind speed were measured in situ with a portable anemometer (Testo 480). DO, pH, ORP, conductivity, and water temperature were measured in situ with portable field probes (Hach HQ40d). Annual air temperature and precipitation were obtained from the National Meteorological Information Center (http://data.cma.cn/).

**Flux computation**. Total $F_{N_2O}$ were calculated according to the equation below:

$$F_t = \frac{n_t \cdot n_0}{A \times t} \qquad (1)$$

where, $n_t$ and $n_0$ are the number of moles of $N_2O$ in the chamber at time t and time zero (mol), respectively; $A$ is the surface area of water covered by the chamber (m²) and $t$ is the measurement duration time (min). Diffusive and ebullitive $N_2O$ fluxes were separated using the Campeau et al. approach[50]. Briefly, we assumed that $F_t$ for $CO_2$ ($F_{CO2}$) is exclusively diffusive (that is, $CO_2$ ebullition is negligible). $F_{CO2}$ was computed from the linear regression of $pCO_2$ against time to eliminate possible bias due to gas accumulation in the chamber headspace that can affect the flux rates. We then used $F_{CO2}$ to calculate $k_{CO2}$ by rearranging the equation for Fick's law of gas diffusion[51]:

$$k_{CO2} = F_{CO2}/(K_H \Delta pCO_2) \qquad (2)$$

where $K_H$ is the temperature-adjusted Henry's constant, and $\Delta pCO_2$ is the $CO_2$ partial pressure difference between surface water and the atmosphere. Finally, the theoretical diffusive $k$ for $N_2O$ was calculated based upon $k_{CO2}$ as follows[52]:

$$k_{N2O}/k_{CO2} = (Sc_{N2O}/Sc_{CO2})^{-n} \qquad (3)$$

where $k_{N2O}$ and $k_{CO2}$ are gas transfer velocity of $N_2O$ and $CO_2$, respectively; $Sc$ is the Schmidt number. $n = 1/2$ is used for rippled or turbulent surface water conditions, or $n = 2/3$ is used for a smooth water surface[53]. We then calculated the theoretical diffusive $F_{N_2O}$ according to

$$F_d = k(C_{water} - C_{eq}) \qquad (4)$$

where $k$ is gas transfer velocity (m·d⁻¹), $C_{water}$ is water gas concentration (mol m⁻³), and $C_{eq}$ is gas concentration in water in equilibrium with the local atmosphere corrected for temperature-induced changes in solubility according to the Henry's law (mol m⁻³). Thus, the difference between the total and diffusive $N_2O$ fluxes is attributable to ebullition.

**Physicochemical analyses**. Duplicate filtered water samples were analyzed for $NH_4^+$-N, $NO_2^-$-N, and $NO_3^-$-N, as well as dissolved organic carbon (DOC) following detailed methods described in ref. [11]. Total phosphorus (TP) was measured on a UV − Vis spectrophotometer (Agilent 1200) using the standard molybdenum blue method after persulfate digestion. Determination of sedimentary $N_2O$ production rates is described in Supplementary Methods 2.

**Microbial analyses**. A total of 26 sediment samples collected across the four rivers between 2016 and 2018 were prepared in triplicate. Genomic DNA was extracted

from approximately 0.5 g fresh homogenized sediment using the FastDNA® SPIN Kit for Soil (MP Biomedicals), following the manufacturer's instructions. Abundances of dissimilatory nitrite reductase (*nirS* and *nirK*)[31] and nitrous oxide reductase (*nosZ*$_I$ and nos$Z_{II}$) genes[32] in these river sediments were estimated by real-time quantitative PCR. Detailed information is available in Supplementary Methods 3.

**GIS analyses.** Vegetation types in the study area are alpine meadow and steppe, and vegetation coverage for each type was extracted using datasets from ref. [54]. the vegetation coverage for each river reach was determined within a circular area of 5 km in radius centered around each sampling site using the ArcGIS 10.6 buffer analysis tool. Normalized difference vegetation index (NDVI) data were obtained from the China Monthly Vegetation Index Spatial Distribution Data Set (Resource and Environment Science and Data Center, Chinese Academy of Sciences, https://www.resdc.cn/), with a spatial resolution of $1 \times 1$ km$^2$ for the period 2016–2018 corresponding to sampling dates. The NDVI values for each river reach were calculated as the mean value within a circular area of 5 km in radius centered around each sampling site using the ArcGIS 10.6 zonal statistics tool.

Hydrological analysis was performed on the ArcGIS 10.4 platform. Streams and rivers were extracted and Strahler stream order for all stream lines in the four catchments was calculated based on Digital Elevation Model (DEM) data. We then calculated the total surface area for each stream order by multiplying the total length with the average width of this stream order. The total length was derived from the distance tool in the ArcGIS. We calculated the average river width for each stream order based on in situ width measurements at the sampling location and 50 additional locations of the corresponding stream order from Google Earth Maps in each catchment. Average stream width for the 1st- and 2nd-order streams were obtained from the width ratios found for rivers of high stream orders according to our earlier work[11].

**Upscaling technique.** We upscaled the magnitude of N$_2$O emissions (and uncertainty) from EQTP streams and rivers with a Monte Carlo simulation (MATLAB R2018b) that ran 1000 iterations for each sampled 3rd- to 7th-stream order. Each iteration randomly selected a N$_2$O flux measurement and simultaneously selected a surface area based on a normal distribution surrounding the mean and standard deviation for that stream in order to generate an order-specific N$_2$O flux per unit of time. This value was then summed across the ice-free season (210 d from April to October) to estimate N$_2$O emission for each stream order. For 1st- and 2nd-order streams, a range of N$_2$O fluxes that extrapolated from the relationship in Fig. 3c and the surface area for this order were used to produce total fluxes and constrain the uncertainty. We summed up the final distribution of N$_2$O emissions from stream orders 1–7, including the mean value and 95% confidence intervals. Finally, we divided the ice-free season emission by 85% to obtain the annual emission. This is because early spring efflux of winter-derived N$_2$O fuels ~15% of the annual emissions[39].

**Statistical analyses.** Pearson correlations and multiple linear stepwise regression analysis were conducted under the Statistical Product and Service Solutions 25.0 software (SPSS) at a significance level of $P < 0.05$. To uncover complex dependencies among predictor variables, we used a regression tree approach to analyze predictors of N$_2$O concentrations with MATLAB R2018b. This nonparametric method does not require linear relationships and allows for interaction effects among predictors. All data form a single group at the top of the tree, and the tree is grown by repeatedly splitting the data into two subgroups, with each split based on the explanatory variable which makes the resulting groups as different as possible. All values of predictive variables are assessed each time a dichotomy is made[55]. For each of the two subgroups, the same process was repeated until a tree is formed. Such a tree needed to be pruned to avoid overfitting and improve predictive accuracy. We then applied 10-fold cross-validation and pruned the tree based on the '1-SE' rule. This was a parsimonious approach to find the smallest tree whose cross-validated relative error (CVRE) is within one standard error of the minimum. We report the amount of variance explained as R$^2$ to depict the fit of the tree. We also report the CVRE since it is a suitable measure of prediction[56]. Predictive variables available for entry into the model were: %O$_2$ and NO$_3$-. Given the high CVRE here, the model is more suggestive than predictive.

**Reporting summary.** Further information on research design is available in the Nature Research Reporting Summary linked to this article.

## Data availability

The data used in this study are available at the Environmental Data Initiative (https://portal.edirepository.org/nis/mapbrowse?packageid=edi.695.1), https://doi.org/10.6073/pasta/ba9340800403c450e7d942d450237dc4.

## Code availability

The Monte Carlo model and regression tree analysis code used in this study are available at the Environmental Data Initiative (https://portal.edirepository.org/nis/mapbrowse?packageid=edi.695.1), https://doi.org/10.6073/pasta/ba9340800403c450e7d942d450237dc4

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

## Acknowledgements

We thank Lin Zhao and Zhiwei Wang at Northwest Institute of Eco-Environment and Resources, CAS who have graciously shared their GIS data to make our vegetation analysis possible. This research was funded by the National Natural Science Foundation of China (grant no. 92047303, 52039001) and the National Key R&D Program of China (grant no. 2017YFA0605001).

## Author contributions

X.X. and L.Z. designed the research. L.Z. and S.Z. collected and analyzed samples. Q.W. and R.L. conducted GIS, Monte Carlo, and regression tree analyses. X.X., L.Z., S.Z., T.J.B., S.L., Z.Y., J.N., and E.H.S. performed data analysis and wrote the manuscript with significant contribution from all authors.

## Competing interests

The authors declare no competing interests.
