## [Peer Review File · Nature Communications]

Title: Unexpectedly minor nitrous oxide emissions from fluvial networks draining permafrost catchments of the East Qinghai-Tibet PlateauREVIEWER COMMENTS

Reviewer #1 (Remarks to the Author):

Review of Zhang et al. Unexpectedly minor nitrous oxide emissions from fluvial networks draining large permafrost catchments of the East Qinghai-Tibet Plateau.

This paper tackles an important topic – seeking to better understand and quantify N₂O emission from streams and rivers, which is a sizable global flux. This study focuses on fluvial networks of the East Qinghai-Tibet Plateau, an area where the extent and pattern in riverine N₂O emissions are largely unknown. The authors measured N₂O emissions in a number of streams and rivers and also conducted microbial analyses and lab incubations to support their field measurements. The team has published another paper on CH₄ emissions from the same area (ref 10 in the manuscript) and I believe the N₂O data is derived from the same field campaigns.

The paper concludes that N₂O emissions from fluvial networks of the East Qinghai-Tibet Plateau are low and attribute this result to low N availability and efficient denitrification in these systems.

Overall, the paper is easy to follow and pleasant to read. The study appears well designed and produces an interesting dataset that will advance our knowledge on riverine N₂O emissions.

I do have a number of concerns about some of the methods and how some conclusions are supported by the data.

First of all, this data is based on 114 measurements of N₂O collected at 26 sites between Spring 2016 and Fall 2018. This is a relatively limited dataset considering the extent of the area but it is easy to understand that sampling in this region is probably challenging. However, I think the number of samples should be clearly reported, for example L 78, by adding n=XX. To find info on the number of samples I had to download the N₂O dataset. But I want to stress that although limited this dataset is valuable.

I do find surprising that several chamber measurements indicate negative N₂O fluxes (uptake of N₂O) while this is not supported by the dissolved concentrations. The authors hypothesize that “measured N₂O reflect transient concentrations, but N₂O consumption through complete denitrification may occur during 60-min floating chamber deployments”. Could it also be due to the fact that the N₂O concentrations were measured at a different spot than the chamber measurements? Were the concentrations measured close to the shore or in the mid channel? How does the relationship between N₂O saturation and chamber-based fluxes look like? It could be nice to show that relationship in supplement. How long were the N₂O samples stored before analysis?

I find the paragraph “Terrestrial processes modulating N₂O dynamics” quite speculative. The authors found a U-shape relationship in their data L 104-105 “N₂O fluxes, together with dissolved N₂O and DIN concentrations, exhibited a U-shaped relationship with permafrost wetland fraction in the catchment”. The authors develop an interesting hypothesis linking plant cover, plant uptake of permafrost-derived N and N₂O emissions. However, the U-shaped relationships, in spite of their impressive R² are only based on 5 points. What I can see from these figures is that higher DIN seems to correlate with higher N₂O concentration and fluxes. To build these U-shaped relationship seems a bit of a stretch with the present dataset. I would therefore encourage the authors to revise this section.

Other comments:

L205 which numbers are underestimated? Stream area or N₂O estimates (or both?)

L258-262. I do not understand how the fluxes were measured. Were the chambers anchored or drifting?

If drifting, how did you keep the chambers in position on the transect?

L277-278 is that the correct reference? I could not find info on separating diffusion and ebullition in this reference.

L283-284 “Sc is the Schmidt number and n is assigned a value of 1/2 for wind speed > 3.6 m s⁻¹ or 2/3 for wind speed < 3.6 m s⁻¹.”

The n is dependent on the surface state of the water. In most streams and rivers, I would assume that the surface state and turbulences are rather related to the slope than to the wind speed. Why did you use wind speed?

Supp Table 1. Discharge: is that mean discharge over the study period? Were there large changes in discharge across seasons? That could increase/decrease turbulences in the river and affect the fluxes.

Supp Table 2 the four headwater catchments could be changed to 5 since Lower and Upper Yellow river are shown as 2 regions/catchments.

Reviewer #2 (Remarks to the Author):

Review report on manuscript by Zhang et al., titled ‘Unexpectedly minor nitrous oxide emissions from fluvial networks draining large permafrost catchments of the East Qinghai-Tibet Plateau’

GENERAL COMMENTS

The present manuscript reports N₂O fluxes and concentrations in a region with a high permafrost coverage in the East Qinghai-Tibet Plateau. Due to the extensive sampling design (4 catchments, ~30 sampling station, samplings during different seasons in all catchments) and novelty (uncertainty of N₂O flux estimates for rivers in general, and complete lack of studies about N₂O fluxes in permafrost region) this is undoubtedly a valuable data set. The research community starts to recognize the importance of nutrient release from permafrost through its direct climatic effects in the form of nitrous oxide emissions to atmosphere and its regulatory role for carbon cycling, and the lateral N export to rivers and its further processing there is a very important piece in this puzzle.

The paper is well written and the methodology suits for the purpose and described with sufficient detail. The dependencies of N₂O fluxes on O₂ saturation and nitrate content (Fig. 2b-c) is summarizing the results of this study very well, and represents a significant advance in the understanding of N biogeochemistry in such permafrost dominated catchments. Although I would like to see this data published, there are major short-comings in the reasoning and conclusion that need careful attention from the authors before I can recommend publication.

MAJOR COMMENTS

1. Acknowledging and discussing the effect of anthropogenic N input.

Throughout the manuscript the authors emphasize the role of permafrost (PF) as an additional source of N to rivers and streams, but they do not comment at all how substantial is the anthropogenic N input to these systems and how does it differ between the rivers. While in the arctic permafrost region the anthropogenic effect is mostly minimal here it is probably not the case. Xia et al. (2019; <https://doi.org/10.1016/j.scitotenv.2019.06.204>) study two of the rivers investigated also here, and estimate that from riverine nitrate 47% is derived from manure and 9% from synthetic fertilizer vs. the contribution of soil organic N of 30% (see lines 113-117). Since the anthropogenic N input most likely varies between the different catchments, any discussion about permafrost derived N is meaningless unless the possible anthropogenic impact is taken into account.

Following improvements are needed:

I) describing the anthropogenic impact in the different catchments (human population, livestock, other sources...) in the main text, methods section and Supplementary Table 2;

II) taking into account and thoroughly discussing the anthropogenic impact on those polynomial dependencies in Fig. 2a, Supplementary Fig. 3a, b;

III) when analyzing the effects of PF on N₂O fluxes, it would be good to exclude any catchments or stations (lower Yellow River?) where the anthropogenic N inputs are significantly higher than in the other stations. Otherwise, these two different signals are mixed which may lead to wrong conclusions about PF effects. If it is impossible to separate the anthropogenic effects from PF thaw effects, the manuscript requires substantial rewriting so that the conclusions are sound.

2. Problems in using the overall wetland coverage as an indicator of PF thaw.

In the text starting from line 103 and in the Fig. 2a wetland coverage is used as a proxy for permafrost thaw. Although it is true that the permafrost thaw and associated ground collapse in certain conditions leads to formation of thermokarst wetlands, the %-coverage used here seems to include all kinds of wetlands. It seems so according to the map in Fig. 1, where the Yellow river catchment has a high wetland coverage, but those wetlands are disconnected from permafrost (although classified as permafrost wetlands in the legend). Also, in the methods section it is stated that: "Permafrost wetlands, defined here as shallow waterlogged habitats, including glacial/post-glacial water bodies, thermokarst water bodies, and peatlands (bogs and fens), have a total surface area of 3.5×10^4 km² on the EQTP". The total coverage of these wetlands of very variable origin cannot be used as a proxy for permafrost thaw, but the authors should a better indicator, such as the total PF coverage/proportion of area underlain with continuous permafrost or else. Also, the term permafrost wetlands is misleading and should not be used when referring to the total wetland coverage.

3. Using previously published data on DOC concentrations and CH₄ fluxes for thorough understanding for riverine biogeochemistry.

As the authors state on lines 150-152, carbon availability may have strong consequences for N₂O production and consumption processes, but they do not discuss the C & N interactions in the manuscript. This should be done, and that should be feasible based on the data published earlier by Zhang et al. (2020; <https://doi.org/10.1038/s41561-020-0571-8>). Besides the possible C limitation of denitrification, DOC mineralization is a key oxygen consuming process and thus very likely an important player in the dependencies shown in Fig. 2 and 3. It would also be great to see how CH₄ and N₂O dynamics are coupled within these rivers, and since the data is available, this unique opportunity should be used.

4. Selecting the right context for the observed flux rates.

The main conclusion of this paper seems to be that the N₂O fluxes from these permafrost dominated catchments are very low. However, I am not totally convinced that the global means are suitable reference data for these fluxes. Fluxes from rivers receiving minor anthropogenic inputs of N should not be compared with rivers heavily impacted by e.g. agriculture. This idea should be kept in mind throughout the MS, but particularly in section "Variability of N₂O concentrations and fluxes" and "Regional and global implications", as well as in Supplementary Table 3 (divide to pristine/intermediate/heavily impacted rivers, and revise the related discussion accordingly). Additionally, the recent synthesis of N₂O emissions from PF soils by Voigt et al. (2020) provides good reference data for these riverine fluxes. It includes also plenty of data from the same region where this study was conducted. Compared to those fluxes, the riverine efflux of N₂O reported here is not that small, actually. The same concerns with nitrate and inorganic concentrations, there some data is available from permafrost regions, and those data should be used for comparison here. After the more appropriate comparison, reconsideration of the title might be needed.

5. Better discussion of N biogeochemistry within rivers.

The Fig. 4 is in my opinion beyond the focus of this paper, which deals with N₂O emissions from rivers, not the terrestrial N discharge. The discussion about the vegetation effects on lines 108-> remains speculative, and is not well connected with the data. Instead, I recommend adding more discussion of the alternative fates after N has entered to river: mineralization in water column/sediment, nitrification in water column/sediment, denitrification and DNRA as an alternative pathway of nitrate. The ammonium data should be reported, and its dependence with nitrate analyzed. If DON is available, showing that data would be also of great value. All in all, the observations here should be better connected to the previous knowledge about the concentrations and fate of N in high-latitude rivers (e.g. as reviewed by Vonk et al. 2015; [doi:10.5194/bg-12-7129-2015](https://doi.org/10.5194/bg-12-7129-2015)). I recommend emphasizing the river processes in the schematic figure Fig. 4 instead of what happens on land, which is out of the scope of this work.

6. Revising the future outlook.

The section 'Future fluvial N₂O emissions under warming climate' is not well connected to the findings of this study and remains vague. Particularly the final sentence is not justified by the data. The data does not really reveal signs of increased N₂O as a results of PF thaw, the opposite, it shows that the emissions factor for nitrate conversion to N₂O is particularly low and the data analysis in the current form does not reveal clear dependencies between PF thaw and nitrate or N₂O flux. I would rather like to see a statement of what we learned from this study, and which gaps should be addressed next.

MINOR COMMENTS

line 25: Talking about three years is overstating, because according to the supplementary table 1, only one of the catchments was sampled over three years. The samplings during different seasons should be rather emphasized since it concerns all the studied rivers.

line 29: It is not clear where the statement: "little N input from terrestrial environments" refers, please revise to match better with the parameters you actually measured here.

lines 35-37: This final statement is not supported by the data and should be removed.

lines 40-41: Add reference!

line 47: How much is the C pool, please quantify&compare to the global pools!

line 53: What is the current knowledge about N pool in PF soils?

line 53: It is a clear overstatement that in PF-affected soils in general are N₂O hotspots. The variability is huge, and still in most of the vegetated, undisturbed PF soils the N cycle is well closed and N₂O fluxes are small or negligible. What has been observed recently, however, that this is not always the case but there are situations where high N₂O emissions occur from PF soils, and the emissions may increase with accelerating PF thaw. Please revise to match with the recent synthesis by Voigt et al.

line 59: instead of spatiotemporal I ask you to be more specific (sampling during different seasons, etc.)

line 62: I do not see mean annual temperature and permafrost coverage anywhere, this is very relevant and should be added

line 70-75: This summary does not fit here in my opinion, but it would be better to go straight to the results.

line 89: short explanation would be good to have here in the main text

line 96: I do not think it is likely that N₂O and CH₄ production would overlap – presence of nitrate should favor N₂O production at the expense of energetically less favorable methanogenesis. Please revise considering this.

line 107: “different processes at work in low and high permafrost catchments” is vague and needs revision

line 110-111: Some clarification is needed with respect to the role of vegetation. Based on the text here it sounds like most of the ground is bare, and the increase in vegetation cover would decrease N discharge to rivers. If this is the case, instead of coverage of alpine vegetation in km² in Supplementray Table 2, it would be better to show the proportion of bare area in %. Or do you claim that alpine vegetation is better than other vegetation types in catching N from soils? Clarification is needed, and the argumentation should be better supported by the data.

lines 132-133: Sounds logical, but how about the role of sediments for N₂O production? See Repert et al. (2014; 10.1002/2014JG002707), who report poor correlation between N₂O production in bed sediments with nitrate concentrations in the river water.

lines 141-142: Stemming from the results of this study, do you have any suggestions of alternative estimation method? What would be need to develop that?

lines 150-152: Only yellow river was used here - the warmest catchment with least permafrost and probably highest anthropogenic effect. How typical this is for all the rivers? Please be open about it and discuss how representative this is for the other rivers.

lines 172-174: Please explain briefly the basic principle of river orders - which order represents the headwaters, which estuary? Also, brief elaboration of “perimeter-to-surface-area ratio and hyporheic exchange rates” would be good to have for permafrost community beyond the limnologists.

line 258: The correlation between N₂O flux and N₂O concentration, please add a scatter plot.

Fig. 1. The river names indicated with orange text are very poorly readable, please place them outside the map or add some background for better readability!

Response to the reviewers' comments

Authors' responses to the reviewers' comments to the manuscript entitled "Unexpectedly minor nitrous oxide emissions from fluvial networks draining large permafrost catchments of the East Qinghai-Tibet Plateau".

We appreciate the insightful comments and suggestions from the reviewers; the comments are extremely helpful for improving the impact and clarity of this manuscript. We have fully discussed the comments and suggestions, and revised the manuscript thoroughly. The detailed revisions are described in point-by-point responses in blue text. Reviewers' comments are labeled as the reviewer and comment number (e.g., **R1-C1** is the first comment by Reviewer 1).

Response to Reviewer #1

Reviewer #1 (Remarks to the Author):

General comments:

R1-C1:

Review of Zhang et al. Unexpectedly minor nitrous oxide emissions from fluvial networks draining large permafrost catchments of the East Qinghai-Tibet Plateau.

This paper tackles an important topic – seeking to better understand and quantify N₂O emission from streams and rivers, which is a sizable global flux. This study focuses on fluvial networks of the East Qinghai-Tibet Plateau, an area where the extent and pattern in riverine N₂O emissions are largely unknown. The authors measured N₂O emissions in a number of streams and rivers and also conducted microbial analyses and lab incubations to support their field measurements. The team has published another paper on CH₄ emissions from the same area (ref. 10 in the manuscript) and I believe the N₂O data is derived from the same field campaigns.

The paper concludes that N₂O emissions from fluvial networks of the East Qinghai-Tibet Plateau are low and attribute this result to low N availability and efficient denitrification in these systems. Overall, the paper is easy to follow and pleasant to read. The study appears well designed and produces an interesting dataset that will advance our knowledge on riverine N₂O emissions.

Response:

We are grateful for the overall positive impression that this reviewer has on our study! These constructive comments are very helpful for improving this manuscript!

R1-C2:

I do have a number of concerns about some of the methods and how some conclusions are supported by the data.

First of all, this data is based on 114 measurements of N₂O collected at 26 sites between Spring 2016 and Fall 2018. This is a relatively limited dataset considering the extent of the area but it is easy to understand that sampling in this region is probably challenging. However, I think the number of samples should be clearly reported, for example L 78, by adding n = XX. To find info on the number of samples I had to download the N₂O dataset. But I want to stress that although limited this dataset

is valuable.

I do find surprising that several chamber measurements indicate negative N₂O fluxes (uptake of N₂O) while this is not supported by the dissolved concentrations. The authors hypothesize that “measured N₂O reflect transient concentrations, but N₂O consumption through complete denitrification may occur during 60-min floating chamber deployments”. Could it also be due to the fact that the N₂O concentrations were measured at a different spot than the chamber measurements? Were the concentrations measured close to the shore or in the mid channel? How does the relationship between N₂O saturation and chamber-based fluxes look like? It could be nice to show that relationship in supplement. How long were the N₂O samples stored before analysis?

Response:

We understand the reviewer’s concern. In fact, we collected triplicate gas samples for dissolved N₂O determination (that is, $114 \times 3 = 342$), and duplicate water samples for NH₄⁺-N and NO_{2,3}⁻-N ($114 \times 2 = 228$) at each site/transect during each sampling campaign as stated in Methods (L280 and L328). The standard deviations of these replicates were usually small, so we presented their average values (n=114) for each site on each unique sample date for data analyses and figures. In the revised manuscript, we have transferred these averages to all plots to better exhibit the spread of data (Fig. 3a, b), and have clarified the text to indicate the number of unique site visits (n=114) and replicate samples (n = 342 for N₂O concentrations, n = 436 for fluxes) that were analyzed (L83-84 and L92-93). Here, n = 436 not $4 \times 114 = 456$, because sometimes less than 4 flux measurements were made at several sites.

Samples for dissolved N₂O concentrations were collected in 3 different locations in each site/transect: 2 locations were next to 2 chambers at near-bank and mid-channel spots, respectively, and 1 location was in the mid-point of the other two sampling spots (see schematic diagram below). Based on our multi-point dissolved N₂O measurements alongside the chamber deployments, as well as results of low N₂O yield, small ratio of *nir/nos*, and lab incubation, we thus proposed the hypothesis highlighted above in the reviewer’s comment. The same phenomenon was also observed in the Upper Mara River in Kenya [Mwanake et al., 2019]. We have added the relationship between N₂O saturation and chamber-based fluxes to Supplementary Fig. 5, which is statistically significant but very weak ($P = 0.026$, $R^2 = 0.01$).

Regarding the storage of dissolved N₂O samples, they were sent back to the lab in Beijing

whenever we reached a city with express delivery. Generally, it took 3-5 days from sampling to analysis in the lab. Our dissolved N_2O concentrations were comparable with the same or neighboring rivers on the Qinghai-Tibet Plateau reported by [Qu et al., 2017] and [Ye et al., 2019], so we are confident in our measurements of N_2O .

Mwanake, R. M., G. M. Gettel, K. S. Aho, D. W. Namwaya, F. O. Masese, K. Butterbach-Bahl, and P. A. Raymond (2019), Land use, not stream order, controls N_2O concentration and flux in the Upper Mara River Basin, Kenya, *Journal of Geophysical Research: Biogeosciences*, 124(11), 3491-3506.

Qu, B., K. S. Aho, C. Li, S. Kang, M. Sillanpää, F. Yan, and P. A. Raymond (2017), Greenhouse gases emissions in rivers of the Tibetan Plateau, *Scientific Reports*, 7(1), 16573.

Ye, R. et al. Concentrations and emissions of dissolved CH_4 and N_2O in the Yarlung Tsangpo River (in Chinese). *Chinese Journal of Ecology* 38, 791-798 (2019).

R1-C3:

I find the paragraph “Terrestrial processes modulating N_2O dynamics” quite speculative. The authors found a U-shape relationship in their data L 104-105 “ N_2O fluxes, together with dissolved N_2O and DIN concentrations, exhibited a U-shaped relationship with permafrost wetland fraction in the catchment”. The authors develop an interesting hypothesis linking plant cover, plant uptake of permafrost-derived N and N_2O emissions. However, the U-shaped relationships, in spite of their impressive R^2 are only based on 5 points. What I can see from these figures is that higher DIN seems to correlate with higher N_2O concentrations and fluxes. To build these U-shaped relationships seems a bit of a stretch with the present dataset. I would therefore encourage the authors to revise this section.

Response:

We thank the reviewer for pointing out the concern about hypothesis linking plant cover, plant uptake of terrestrial N, and riverine N₂O in the manuscript. Combined with the other reviewer's comments, we have deleted the U-shape figure and thoroughly rewritten this section (L109-133), and do hope that this revision makes a more compelling case. We describe some of these changes here, but please also see the response to R2-C2 for a detailed discussion of these updates. Additionally, we added relevant Methods (L275-278 and L341-349), as well as a new paragraph to the **Supplementary Discussion 2**.

A recent study documented elevated terrestrial N uptake by plants across the Qinghai-Tibet Plateau [Kou *et al.*, 2020]. At the same time, another study provided evidence that soil-derived NH₄⁺ and NO₃⁻ were detected in Arctic tundra plant tissues, and these plants took up soil NO₃⁻ at comparable rates to plants from relatively NO₃⁻-rich ecosystems in tropical and subtropical biomes [Liu *et al.*, 2018]. We thus assume that an increase in vegetation cover results in greater plant uptake of terrestrial N, meaning that some fraction of terrestrial N can be sequestered instead of entering rivers. In this revision, **we added new data to examine the role of vegetation coverage (for all sites) and NDVI (for sites in non-permafrost zone) in reducing riverine DIN in order to provide a more rigorous examination of the 'more plants, less riverine DIN' argument.**

Regarding the relationship between DIN and N₂O, DIN indeed had a positive correlation with N₂O concentrations. However, NH₄⁺ showed significant but weak correlation with N₂O concentrations ($P < 0.001$, $R^2 = 0.1$) as shown in **Supplementary Table 4**. Apparently, NH₄⁺ was a minor driver of N₂O concentrations as stated in L135-137 in the main text. So, we discussed the relationship of NO₃⁻ and N₂O concentrations in detail in next section — “Biogeochemical processes regulating N₂O dynamics”.

Kou, D., et al. (2020), Progressive nitrogen limitation across the Tibetan alpine permafrost region, *Nature Communications*, 11(1), 3331.

Liu, X., et al. (2018), Nitrate is an important nitrogen source for Arctic tundra plants, *Proceedings of the National Academy of Sciences*, 115(13), 3398-3403.

Other comments:

R1-C4: L205. Which numbers are underestimated? Stream area or N₂O estimates (or both?)

Response:

We have revised the sentence as below:

“these N₂O estimates are probably overestimated.” (L225)

R1-C5: L258-262. I do not understand how the fluxes were measured. Were the chambers anchored or drifting? If drifting, how did you keep the chambers in position on the transect?

Response:

The chambers were anchored. The sentence has been modified to make it clearer as follows:

“Four floating chambers were held in place at each transect.” (L290)

R1-C6: L277-278. Is that the correct reference? I could not find info on separating diffusion and ebullition in this reference.

Response:

We have replaced the correct reference: Regional contribution of CO₂ and CH₄ fluxes from the fluvial network in a lowland boreal landscape of Québec. (L308-309)

R1-C7: L283-284. “Sc is the Schmidt number and n is assigned a value of 1/2 for wind speed > 3.6 m s⁻¹ or 2/3 for wind speed < 3.6 m s⁻¹.” The n is dependent on the surface state of the water. In most streams and rivers, I would assume that the surface state and turbulences are rather related to the slope than to the wind speed. Why did you use wind speed?

Response:

We have clarified these methods by further explaining how the gas transfer velocity (k) for N₂O can be calculated from direct measurements of CO₂ flux. These methods are well established and widely used to estimate gas fluxes, including the choice of exponent values for the ratios of Schmidt numbers for CO₂ and N₂O. The reviewer is correct that the key feature is surface turbulence in rivers. These calculations originated in pelagic systems where surface turbulence is a function of wind speed, and thus the language used to describe these equations and exponents often reflects this origin. Regardless of the source of turbulence (wind vs river channel features), the critical issue is the degree of surface water smoothness. Thus, we re-worded the text to state that use of n = 2/3 vs. n =

1/2 was in fact based on the degree of surface turbulence (L312-321).

R1-C8: Supp Table 1. Discharge: is that mean discharge over the study period? Were there large changes in discharge across seasons? That could increase/decrease turbulences in the river and affect the fluxes.

Response:

Yes, these values are the mean discharge values for each sampling site over the study period. Indeed, discharge at some sites displayed large variation across seasons. Although we did find a significant relationship between discharge and N₂O flux (see figure below), it was extremely weak ($R^2 = 0.01$ and a slope near 0). Therefore, we did not focus on the effect of discharge on N₂O fluxes.

Supplementary Fig. 2 shows seasonal patterns in N₂O concentrations and fluxes, and discussion about seasonal dynamics is presented in L86-87, L94-97 in the main text and Supplementary Discussion 1.

R1-C9: Supp Table 2. The four headwater catchments could be changed to 5 since Lower and Upper Yellow River are shown as 2 regions/catchments.

Response:

Thanks, we have changed to 5 catchments, and added a brief explanation to Supplementary Table 2 caption as to why we separated the Yellow River into upper and lower sections.

Response to Reviewer #2

Reviewer #2 (Remarks to the Author):

General comments:

R2-C1:

Review report on manuscript by Zhang et al., titled “Unexpectedly minor nitrous oxide emissions from fluvial networks draining large permafrost catchments of the East Qinghai-Tibet Plateau”.

The present manuscript reports N₂O fluxes and concentrations in a region with a high permafrost coverage in the East Qinghai-Tibet Plateau. Due to the extensive sampling design (4 catchments, ~30 sampling station, samplings during different seasons in all catchments) and novelty (uncertainty of N₂O flux estimates for rivers in general, and complete lack of studies about N₂O fluxes in permafrost region), this is undoubtedly a valuable data set. The research community starts to recognize the importance of nutrient release from permafrost through its direct climatic effects in the form of nitrous oxide emissions to atmosphere and its regulatory role for carbon cycling, and the lateral N export to rivers and its further processing there is a very important piece in this puzzle.

The paper is well written and the methodology suits for the purpose and described with sufficient detail. The dependencies of N₂O fluxes on O₂ saturation and nitrate content (Fig. 2b-c) is summarizing the results of this study very well, and represents a significant advance in the understanding of N biogeochemistry in such permafrost dominated catchments. Although I would like to see this data published, there are major short-comings in the reasoning and conclusion that need careful attention from the authors before I can recommend publication.

Response:

We thank this reviewer for the positive comments, which are very helpful in producing a stronger manuscript!

Because some comments are interrelated, we would like to make some clarifications first. Before writing this manuscript, we determined that N₂O concentrations, fluxes, and the magnitude of emissions from East Qinghai-Tibet Plateau (EQTP) rivers were low (see detailed explanations in R2-C5 below). For insurance purposes, we also consulted three experts. This led us to establish the following points to organize the “Results and Discussion” by first describing the basic patterns in N₂O concentrations and fluxes, then explaining why N₂O was low:

1. The current limited knowledge tells us that rivers are sites of net N processing instead of N production, owing to limited N₂ fixation [Marcarelli *et al.*, 2008] relative to inputs from land, intermediate runoff, and groundwater. Further, some fraction of terrestrial N from both natural and anthropogenic sources can be taken up by terrestrial plants instead of entering rivers, hence reducing N loads to EQTP streams and rivers.

- Key supporting result: Negative relationships between either vegetation cover (for all sites) or NDVI (for sites in non-continuous permafrost area) and riverine DIN (new Fig. 2).

2. Well oxygenated overlying water of EQTP rivers limits the extent of hypoxic-anoxic conditions needed for N₂O generation via denitrification.

- Key supporting results: Positive relationship between NO₃⁻ and N₂O when %O₂ < 100% (n = 88) vs. relatively flat relationship between NO₃⁻ and N₂O when %O₂ ≥ 100% (n = 139; Fig. 3a now); Mean EF_{5-r} for EQTP rivers is low (L151-159).

3. High conversion of N₂O to N₂.

- Key supporting result: Low N₂O yield (L161-169) and small ratio of *nir/nos* (L170-181).

In spite of the small magnitude of N₂O fluxes at present, these alpine permafrost rivers may become strong emitters of N₂O in the coming decades (L240-253):

I. Permafrost is expected to thaw gradually from the surface downwards in the warmer future.

Dissolved N from deep soils that extend beyond rhizospheres will be minimally affected by plant uptake [Kou *et al.*, 2020], hence increasing terrestrial N input into surrounding rivers via deep flow paths. This phenomenon has been observed in high-latitude rivers [Harms and Jones Jr., 2012; Khosh *et al.*, 2017]. (This corresponds to the section of “Terrestrial processes modulating N₂O dynamics”)

II. Denitrification has a higher optimal temperature than anammox. A recent study has indicated that denitrifiers are more thermotolerant, whereas anammox bacteria are relatively psychrotolerant [Tan *et al.*, 2020]. So, warmer water temperatures in these cryospheric rivers in the future will reduce N₂O solubility, and may suppress anammox, and direct more of the N flow towards denitrification and associated N₂O production (i.e., higher N₂O yield due to less N₂ production via anammox). At the same time, increasing water temperature drives widespread declines in DO owing to reduced O₂ solubility. Based on Fig. 3a, b in the main text, hypoxia is good for promoting N₂O production via

denitrification (i.e., higher EF_{5-r} and higher N_2O yield due to less N_2O reduction via denitrification). (This corresponds to the section of “Biogeochemical processes regulating N_2O dynamics”, and discussion on the emission factor ($EF_{5-r} = N_2O-N/NO_3^- -N$); N_2O yield [$\Delta N_2O/(\Delta N_2O + \Delta N_2) \times 100\%$])

III. The majority of the changes in the northern mid- to high latitudes are towards less river ice cover, with the greatest declines around the Qinghai-Tibet Plateau, Siberia, and Alaska [Yang *et al.*, 2020]. The emissions of N_2O accumulated underneath the ice during winter may be very limited in permafrost-affected systems due to minimum N inputs from frozen soils to rivers in winter. However, the extension of the ice-free duration means the annual fluvial N_2O emissions likely have an increasing trend. (This corresponds to L225-229)

According to the above future outlook, we redrew the Fig. 4 (Fig.5 now) to visualize this conceptualization.

Harms, T. K., and J. B. Jones Jr. (2012), Thaw depth determines reaction and transport of inorganic nitrogen in valley bottom permafrost soils, *Global Change Biology*, 18(9), 2958-2968.

Khosh, M. S., J. W. McClelland, A. D. Jacobson, T. A. Douglas, A. J. Barker, and G. O. Lehn (2017), Seasonality of dissolved nitrogen from spring melt to fall freezeup in Alaskan Arctic tundra and mountain streams, *Journal of Geophysical Research: Biogeosciences*, 122(7), 1718-1737.

Kou, D., et al. (2020), Progressive nitrogen limitation across the Tibetan alpine permafrost region, *Nature Communications*, 11(1), 3331.

Marcarelli, A. M., M. A. Baker, and W. A. Wurtsbaugh (2008), Is in-stream N_2 fixation an important N source for benthic communities and stream ecosystems? *Journal of the North American Benthological Society*, 27(1), 186-211.

Tan, E., W. Zou, Z. Zheng, X. Yan, M. Du, T.-C. Hsu, L. Tian, J. J. Middelburg, T. W. Trull, and S.-j. Kao (2020), Warming stimulates sediment denitrification at the expense of anaerobic ammonium oxidation, *Nature Climate Change*, 10(4), 349-355.

Yang, X., T. M. Pavelsky, and G. H. Allen (2020), The past and future of global river ice, *Nature*, 577(7788), 69-73.

Major comments:

R2-C2:

1. Acknowledging and discussing the effect of anthropogenic N input.

Throughout the manuscript the authors emphasize the role of permafrost (PF) as an additional source of N to rivers and streams, but they do not comment at all how substantial is the anthropogenic N input to these systems and how does it differ between the rivers. While in the arctic permafrost region the anthropogenic effect is mostly minimal, here it is probably not the case. Xia et al. (2019; <https://doi.org/10.1016/j.scitotenv.2019.06.204>) study two of the rivers investigated also here, and estimate that from riverine nitrate 47% is derived from manure and 9% from synthetic fertilizer vs. the contribution of soil organic N of 30% (see lines 113-117). Since the anthropogenic N input most likely varies between the different catchments, any discussion about permafrost derived N is meaningless unless the possible anthropogenic impact is taken into account.

Following improvements are needed:

I) describing the anthropogenic impact in the different catchments (human population, livestock, other sources...) in the main text, methods section and Supplementary Table 2;

II) taking into account and thoroughly discussing the anthropogenic impact on those polynomial dependencies in Fig. 2a, Supplementary Fig. 3a, b;

III) when analyzing the effects of PF on N₂O fluxes, it would be good to exclude any catchments or stations (lower Yellow River?) where the anthropogenic N inputs are significantly higher than in the other stations. Otherwise, these two different signals are mixed which may lead to wrong conclusions about PF effects. If it is impossible to separate the anthropogenic effects from PF thaw effects, the manuscript requires substantial rewriting so that the conclusions are sound.

Response:

We thank the reviewer for touching on an important point. We realize that we should take anthropogenic N input into account, so we have noted the growing human influence on the QTP in Introduction (L73-75) and Future outlook (L248-249) sections, and added new data (total human population size and population density) to Supplementary Table 2, and a new sub-section on “anthropogenic N inputs on the QTP” to Supplementary Discussion 2. We have also substantially rewritten the “Terrestrial processes modulating N₂O dynamics” in the main text to **emphasize the role of plants in terrestrial N uptake** in this section. **Both natural and anthropogenic N can be**

taken up by plants, thus a part of terrestrial N can be sequestered before entering rivers. Please see the response to R1-C3 for further details.

Despite differences in population densities and contributions of anthropogenic N inputs among the 4 (Supplementary Fig. 2) or 5 basins (according to Supplementary Table 2, see figures below), we highlight that there were no statistically significant differences in N₂O concentrations and fluxes among these basins (L87-90 and L97-98 in the main text). And here, **N₂O concentrations, N₂O fluxes, and riverine DIN (L130-133) were low even with modest anthropogenic N inputs.**

Besides, we used our own data and [Begum *et al.*, 2021; Wu *et al.*, 2020; Xia *et al.*, 2021] data to compare N₂O concentrations and fluxes in different sections of the whole basins of the Lancang-Mekong and Yellow Rivers (see figures below).

Lancang-Mekong River: headwater basin is located in the Qinghai-Tibet Plateau (population density = 3.7 km⁻²; our own data in Supplementary Table 2), upper basin is located in Yunnan Province (population density = 46 km⁻²), mid-lower basin is located in Myanmar, Laos, Thailand, Cambodia and Vietnam (population density = 101 km⁻²).

Yellow River: upper and lower headwater basins are located in the Qinghai-Tibet Plateau (population density = 1.3 and 8.4 km⁻², respectively; our own data in Supplementary Table 2), upper basin is located in the Loess Plateau (Gansu, Ningxia and Inner Mongolia Provinces; population density = 27 km⁻²), and middle basin is located in the Loess Plateau (Shaanxi and Shanxi Provinces; population density = 209 km⁻²).

We found that N₂O concentrations and fluxes showed increasing trends with increasing population density, even if differences among sub-basins were not statistically significant. These within-basin comparisons suggest that significant differences in N₂O concentrations and fluxes emerge only when the population density has surpassed a threshold of ca. > 100 km⁻².

Begum, M. S., et al. (2021), Localized pollution impacts on greenhouse gas dynamics in three anthropogenically modified Asian river systems, *Journal of Geophysical Research: Biogeosciences*, 126(5), e2020JG006124.

Wu, W., J. Wang, X. Zhou, B. Yuan, M. Guo, and L. Ren (2020), Spatiotemporal distribution of nitrous oxide (N₂O) emissions from cascade reservoirs in Lancang-Mekong River Yunnan section, Southwestern China, *River Research and Applications*, n/a(n/a).

Xia, X., L. Zhang, G. Wang, J. Wang, L. Zhang, S. Zhang, and Z. Li (2021), Nitrogen loss from a turbid river network based on N₂ and N₂O fluxes: Importance of suspended sediment, *Science of The Total Environment*, 757, 143918.

Population densities in the two basins were obtained from

Xu, Y., and C. Wang (2020), Ecological protection and high-quality development in the Yellow River Basin: Framework, path, and countermeasure (in Chinese), *Bulletin of Chinese Academy of Sciences*, (7): 875-883.

You, Z., Z. Feng, L. Jiang, and Y. Yang (2014), Population distribution and its spatial relationship with terrain

elements in Lancang-Mekong River Basin (in Chinese), *Mountain Research*, 32(1): 21-29.

R2-C3:

2. Problems in using the overall wetland coverage as an indicator of PF thaw.

In the text starting from line 103 and in the Fig. 2a wetland coverage is used as a proxy for permafrost thaw. Although it is true that the permafrost thaw and associated ground collapse in certain conditions leads to formation of thermokarst wetlands, the %-coverage used here seems to include all kinds of wetlands. It seems so according to the map in Fig. 1, where the Yellow River catchment has a high wetland coverage, but those wetlands are disconnected from permafrost (although classified as permafrost wetlands in the legend). Also, in the methods section it is stated that: “Permafrost wetlands, defined here as shallow waterlogged habitats, including glacial/post-glacial water bodies, thermokarst water bodies, and peatlands (bogs and fens), have a total surface area of 3.5×10^4 km² on the EQTP”. The total coverage of these wetlands of very variable origin cannot be used as a proxy for permafrost thaw, but the authors should a better indicator, such as the total PF coverage/proportion of area underlain with continuous permafrost or else. Also, the term “permafrost wetlands” is misleading and should not be used when referring to the total wetland coverage.

Response:

Thanks for raising this key point. We recognize that this approach as an indicator of permafrost thaw was ambiguous. Thus, for this revision, we adopted the more straightforward method that was used in [Karlsson *et al.*, 2021; Serikova *et al.*, 2018] to characterize differences in permafrost among sites (see L275-278 in the Method). We considered NDVI vs DIN relationships for these two categories (and found that they were distinct), and eliminated the previous plot of N₂O (and DIN) vs. permafrost wetlands that suggested a polynomial relationship (point II in R2-C2). We are grateful for the reviewer raising this issue, as we feel that this revised analysis provides strong and clear support for distinct relationships between plant productivity/greenness and riverine DIN associated with soil permafrost status.

Karlsson, J., S. Serikova, S. N. Vorobyev, G. Rocher-Ros, B. Denfeld, and O. S. Pokrovsky (2021), Carbon emission from Western Siberian inland waters, *Nature Communications*, 12(1), 825.

Serikova, S., et al. (2018), High riverine CO₂ emissions at the permafrost boundary of Western Siberia, *Nature*

R2-C4:

3. Using previously published data on DOC concentrations and CH₄ fluxes for thorough understanding for riverine biogeochemistry.

As the authors state on lines 150-152, carbon availability may have strong consequences for N₂O production and consumption processes, but they do not discuss the C & N interactions in the manuscript. This should be done, and that should be feasible based on the data published earlier by Zhang et al. (2020; <https://doi.org/10.1038/s41561-020-0571-8>). Besides the possible C limitation of denitrification, DOC mineralization is a key oxygen consuming process and thus very likely an important player in the dependencies shown in Fig. 2 and 3. It would also great to see how CH₄ and N₂O dynamics are coupled within these rivers, and since the data is available, this unique opportunity should be used.

Response:

Thanks for this thoughtful suggestion. We have added a new figure (Fig. 4a) to the main text to show the effect of DOC on N₂O yield.

Actually, we did try to establish some relationships between N and C cycles while writing. However, we only found insignificant or weak relationships ($R^2 < 0.1$) between N and C cycles (see figures below), including DOC, CO₂ and CH₄, so we did not pursue this avenue further.

We did not find any significant relationships between DOC and either DO or %O₂ as well (see figures below), suggesting DOC mineralization does not significantly affect the oxygen status of these rivers.

We put the following variables in the regression tree analysis: %O₂, altitude, air pressure, water temperature, pH, DO, DOC, NH₄⁺, NO₃⁻, DIN, and TP. The best predictive variables available for entry into the tree model included only %O₂ and NO₃⁻, and yielded the model with the highest R² value (R² = 0.56; Fig. 3b). This is indicative of the first and primary role of %O₂ (rather than DOC) in controlling N₂O dynamics.

Finally, we added relevant discussion to the main text (L229-231) as below, and the similar phenomenon was also observed in boreal aquatic networks in Québec [Soued *et al.*, 2016]:

“These alpine permafrost waterways emit large amounts of CH₄, but fortunately they are small contributors of N₂O delivery to the atmosphere, demonstrating CH₄ and N₂O dynamics are uncoupled within these systems.”

Soued, C., P. A. del Giorgio, and R. Maranger (2016), Nitrous oxide sinks and emissions in boreal aquatic networks in Québec, *Nature Geoscience*, 9(2), 116-120.

R2-C5:

4. Selecting the right context for the observed flux rates.

The main conclusion of this paper seems to be that the N₂O fluxes from the permafrost dominated catchments are very low. However, I am not totally convinced that the global means are suitable reference data for these fluxes. Fluxes from rivers receiving minor anthropogenic inputs of N should not be compared with rivers heavily impacted by e.g., agriculture. This idea should be kept in mind throughout the MS, but particularly in section “Variability of N₂O concentrations and fluxes” and “Regional and global implications”, as well as in Supplementary Table 3 (divide to pristine/intermediate/heavily impacted rivers, and revise the related discussion accordingly). Additionally, the recent synthesis of N₂O emissions from PF soils by Voigt et al. (2020) provides good reference data for these riverine fluxes. It includes also plenty of data from the same region where this study was conducted. Compared to those fluxes, the riverine efflux of N₂O reported here is not that small, actually. The same concerns with nitrate and inorganic concentrations, there some data is available from permafrost regions, and those data should be used for comparison here. After the more appropriate comparison, reconsideration of the title might be needed.

Response:

We understand the reviewer’s concern. As stated above, we would like to continue to hold the position that N₂O concentrations, fluxes, and the magnitude of emissions from these permafrost rivers are very low.

Firstly, we did some calculations based on [*Tian et al., 2019*] and [*Voigt et al., 2020*]: Both natural and anthropogenic soils around the world emitted N₂O at a mean rate of 0.07 g N₂O-N m⁻² yr⁻¹ = 192 μg N₂O-N m⁻² d⁻¹ = 6.9 μmol m⁻² d⁻¹, which is 1.5 times lower than permafrost-affected soils (288 μg N₂O-N m⁻² d⁻¹ = 10.3 μmol m⁻² d⁻¹). And 10.3% of upscaled global soil N₂O emission (131.5

$\times 10^6 \text{ km}^2$, $10.0 \text{ Tg N}_2\text{O-N yr}^{-1}$) comes from Northern Hemisphere permafrost regions ($17.8 \times 10^6 \text{ km}^2$, $1.03 \text{ Tg N}_2\text{O-N yr}^{-1}$), which is a proportionate contribution given their areal contribution to global soils (13.5%). So, permafrost soils are evident or even substantial sources of N_2O to the atmosphere. **In contrast**, permafrost rivers draining the EQTP emitted N_2O at a mean rate of $9.1 \mu\text{mol m}^{-2} \text{ d}^{-1}$, which is 10.4 times lower than the world's average ($94.3 \mu\text{mol m}^{-2} \text{ d}^{-1}$) [Hu *et al.*, 2016]. And we obtained a upscaled riverine N_2O emission of $0.432\text{--}0.463 \text{ Gg N}_2\text{O-N yr}^{-1}$ from the entire QTP ($5,141 \text{ km}^2$ of river channel area), which is minor ($\sim 0.15\%$) given their areal contribution (0.7%) to global streams and rivers ($291.3 \text{ Gg N}_2\text{O-N yr}^{-1}$, $773,000 \text{ km}^2$) [Yao *et al.*, 2020]. When normalized to river or basin area, QTP rivers also released the smallest magnitude of N_2O to the atmosphere, which is an order of magnitude lower than those from lotic systems worldwide (L218-225). So, QTP rivers seem to be minor sources of N_2O to the atmosphere.

Secondly, we think riverine N_2O fluxes and the magnitude of emissions should be compared with other rivers rather than soils, even if most reported rivers are heavily impacted by agriculture and urbanization. Rivers and soils are fundamentally different matrixes, and their environmental conditions are very different, such as pH, temperature, redox and oxygen level. At the same time, there are no dissolved N_2O concentrations and N_2O bubble fluxes in soils as far as we know, which makes the comparisons of dissolved N_2O concentrations and N_2O bubble fluxes (ebullition) between rivers and soils impossible. Although most existing studies of fluvial N_2O dynamics focus on human-influenced lowland systems, we were able to add additional results from recent publications to Supplementary Table 3 that include other lotic systems in mountain environments.

Hu, M., D. Chen, and R. A. Dahlgren (2016), Modeling nitrous oxide emission from rivers: A global assessment, *Global Change Biology*, 22(11), 3566-3582.

Tian, H., et al. (2019), Global soil nitrous oxide emissions since the preindustrial era estimated by an ensemble of terrestrial biosphere models: Magnitude, attribution, and uncertainty, *Global Change Biology*, 25(2), 640-659.

Voigt, C., M. E. Marushchak, B. W. Abbott, C. Biasi, B. Elberling, S. D. Siciliano, O. Sonnentag, K. J. Stewart, Y. Yang, and P. J. Martikainen (2020), Nitrous oxide emissions from permafrost-affected soils, *Nature Reviews Earth & Environment*, 1(8), 420-434.

Yao, Y., H. Tian, H. Shi, S. Pan, R. Xu, N. Pan, and J. G. Canadell (2020), Increased global nitrous oxide emissions from streams and rivers in the Anthropocene, *Nature Climate Change*, 10(2), 138-142.

R2-C6:

5. Better discussion of N biogeochemistry within rivers.

The Fig. 4 is in my opinion beyond the focus of this paper, which deals with N₂O emissions from rivers, not the terrestrial N discharge. The discussion about the vegetation effects on lines 108-> remains speculative, and is not well connected with the data. Instead, I recommend adding more discussion of the alternative fates after N has entered to river: mineralization in water column/sediment, nitrification in water column/sediment, denitrification and DNRA as an alternative pathway of nitrate. The ammonium data should be reported, and its dependence with nitrate analyzed. If DON is available, showing that data would be also of great value. All in all, the observations here should be better connected to the previous knowledge about the concentrations and fate of N in high-latitude rivers (e.g., as reviewed by Vonk et al. 2015; doi:10.5194/bg-12-7129-2015). I recommend emphasizing the river processes in the schematic figure Fig. 4 instead of what happens on land, which is out of the scope of this work.

R2-C7:

6. Revising the future outlook.

The section 'Future fluvial N₂O emissions under warming climate' is not well connected to the findings of this study and remains vague. Particularly the final sentence is not justified by the data. The data does not really reveal signs of increased N₂O as a results of PF thaw, the opposite, it shows that the emissions factor for nitrate conversion to N₂O is particularly low and the data analysis in the current form does not reveal clear dependencies between PF thaw and nitrate or N₂O flux. I would rather like to see a statement of what we learned from this study, and which gaps should be addressed next.

Response:

We understand the reviewer's concern. As stated in the response to R2-C1, the future outlook is closely connected to the findings of this study. The final sentence also corresponds to Fig. 3a, b. Based on our future forecast (I) and (II) in the response to R2-C1, more terrestrial N can enter rivers via deep flow paths in the warmer future, then more NO₃⁻ converts to N₂O via denitrification at the expense of annamox under hypoxic-anoxic conditions in the warmer waters (Fig. 3a). This prediction fits our regression tree model well (Fig. 3b), thus likely leading to a positive non-C climate feedback

of currently unanticipated magnitude due to an increase in riverine N₂O production following the development of climate change.

Though the emissions factor ($EF_{5-r} = N_2O-N/NO_3^-N$) is low at present (this result corroborates our finding that well oxygenated overlying water of EQTP rivers limits the extent of hypoxic-anoxic conditions needed for N₂O generation via denitrification), it is expected that widespread hypoxia in future warmer waters may boost N₂O production via denitrification as we predict in R2-C1.

We did not pay attention to DNRA because the contribution of this pathway is very limited in these rivers [Zhang *et al.*, 2021]. There is no relationship between NH₄⁺ and NO₃⁻ (see figure below). At the same time, NH₄⁺ had a significant but weak correlation with N₂O concentrations ($P < 0.001$, $R^2 = 0.1$) as presented in Supplement Table 4. We also did not focus on nitrification, since NH₄⁺ was a minor driver of N₂O concentrations as stated in L135-137 in the main text. Lastly, we regret that we don't have DON data, so we can't discuss mineralization in depth. In contrast, denitrification and anammox play key roles in these rivers [Zhang *et al.*, 2021], so we emphasized these two processes that are closely related to our findings in the future outlook (please see the response to R2-C1 for detail). Our inclination is to kindly underscore the main line of the story instead of being distracted by some insignificant and/or weak relationships.

We have strengthened the ending with more detailed discussion to make a clarification of what we learned from this study, and which gaps should be addressed next (L240-262). We also revised Fig. 5 to pay more attention to river processes.

Zhang, S., W. Qin, Y. Bai, Z. Zhang, J. Wang, H. Gao, J.-D. Gu, and X. Xia (2021), Linkages between anammox and denitrifying bacterial communities and nitrogen loss rates in high-elevation rivers, *Limnology and Oceanography*, 66(3), 765-778.

Minor comment:

R2-C8: line 25. Talking about three years is overstating, because according to the supplementary table 1, only one of the catchments was sampled over three years. The samplings during different seasons should be rather emphasized since it concerns all the studied rivers.

Response:

We reorganized the sentence as “Here we directly measured N₂O concentrations and fluxes during different seasons between 2016 and 2018 in four watersheds on the East Qinghai-Tibet Plateau.”

(L26-28)

R2-C9: line 29. It is not clear where the statement: “little N from terrestrial environments” refers, please revise to match better with the parameters you actually measured here.

Response:

We have modified this sentence as “Such low N₂O fluxes were associated with low riverine DIN after terrestrial plant uptake, unfavorable conditions (usually supersaturated dissolved oxygen) for N₂O generation via denitrification, and low N₂O yield due to small ratio of nitrite reductase: nitrous oxide reductase in these rivers.” (L31-34)

R2-C10: lines 35-37: This final statement is not supported by the data and should be removed.

Response:

As stated in the response to **R2-C1** and **R2-C6/7**, the final sentence is closely connected to our results. So, we would like to kindly keep it as it is. (L36-38)

R2-C11: lines 40-41: Add reference!

Response:

Thanks. We have added the reference to the sentence as “Sources of this powerful GHG are poorly constrained in general, and for global streams and rivers in particular [*Quick et al.*, 2019]” (L41-42)

Quick, A. M., W. J. Reeder, T. B. Farrell, D. Tonina, K. P. Feris, and S. G. Benner (2019), Nitrous oxide from streams and rivers: A review of primary biogeochemical pathways and environmental variables, *Earth-*

Science Reviews, 191, 224-262.

R2-C12: line 47: How much is the C pool, please quantify & compare to the global pools!

Response:

We reorganized this sentence as “Massive amounts of organic carbon (OC, ~1014 Pg C) are stored in the top 3 m of Northern Hemisphere permafrost soils [Mishra *et al.*, 2021].” (L48-49)

Mishra, U., *et al.* (2021), Spatial heterogeneity and environmental predictors of permafrost region soil organic carbon stocks, *Science Advances*, 7(9), eaaz5236.

R2-C13: line 53: What is the current knowledge about N pool in PF soils?

R2-C14: line 53: It is a clear overstatement that in PF-affected soils in general are N₂O hotspots. The variability is huge, and still in most of the vegetated, undisturbed PF soils the N cycle is well closed and N₂O fluxes are small or negligible. What has been observed recently, however, that this is not always the case but there are situations where high N₂O emissions occur from PF soils, and the emissions may increase with accelerating PF thaw. Please revise to match with the recent synthesis by Voigt *et al.*

Response:

We strictly followed the description in [Voigt *et al.*, 2020], and modified the sentence as “Northern Hemisphere permafrost soils contain 67 Pg N to a depth of 3 m (excluding N pools in the active layer), and are evident or even substantial sources of N₂O”. (L54-55)

Voigt, C., M. E. Marushchak, B. W. Abbott, C. Biasi, B. Elberling, S. D. Siciliano, O. Sonnentag, K. J.

Stewart, Y. Yang, and P. J. Martikainen (2020), Nitrous oxide emissions from permafrost-affected soils, *Nature Reviews Earth & Environment*, 1(8), 420-434.

R2-C15: line 59: instead of spatiotemporal I ask you to be more specific (sampling during different seasons, etc.).

Response:

We have changed to “we provide the first cross-regional and seasonal direct measurements of fluvial N₂O concentrations and fluxes.” (L61-62)

R2-C16: line 62: I do not see mean annual temperature and permafrost coverage anywhere, this is very relevant and should be added.

Response:

We have added the info to Supplementary Table 2.

R2-C17: line 70-75: This summary does not fit here in my opinion, but it would be better to go straight to the results.

Response:

We have modified this summary to make it fit better with the discussion of future outlook. (L77-79)

R2-C18: line 89: short explanation would be good to have here in the main text.

Response:

We have revised this sentence as “The asynchronous seasonal patterns between concentrations and fluxes are likely caused by water temperature and precipitation (Supplementary Discussion 1).” (L95-97)

R2-C19: line 96: I do not think it is likely that N₂O and CH₄ production would overlap – presence of nitrate should favor N₂O production at the expense of energetically less favorable methanogenesis. Please revise considering this.

Response:

Sorry for the confusing expression. We have deleted “formation”. (L102-103)

R2-C20: line 107: “different processes at work in low and high permafrost catchments” is vague and needs revision.

R2-C21: line 110-111: Some clarification is needed with respect to the role of vegetation. Based on the text here it sounds like most of the ground is bare, and the increase in vegetation cover would decrease N discharge to rivers. If this is the case, instead of coverage of alpine vegetation in km² in Supplementary Table 2, it would be better to show the proportion of bare area in %. Or do you claim that alpine vegetation is better than other vegetation types in catching N from soils? Clarification is

needed, and the argumentation should be better supported by the data.

Response:

Please see the responses to R1-C3 and R2-C2 above. We have also added a vegetation map to Supplementary Fig. 1, and total vegetation and barren land (km² and %) to Supplementary Table 2.

R2-C22: lines 132-133: Sounds logical, but how about the role of sediments for N₂O production? See Repert et al. (2014; 10.1002/2014JG002707), who report poor correlation between N₂O production in bed sediments with nitrate concentrations in the river water.

Response:

Yes, it is possible that small amounts of N₂O could be steadily delivered to the channels via different flow paths, including benthic or hyporheic sediments. So, we added a sentence “N₂O present at these sites was derived from rare surface sediment patches in the channel that maintain hypoxic-anoxic conditions despite abundant O₂ in the water column” to L144-145 in the main text, because N₂O could also be generated within benthic-hyporheic sediments that have limited hydrologic exchange with the overlying water and thus low O₂, even when overlying O₂ is >100% (e.g., some small deposit of fine sediments, see sediment types in Supplementary Table 1). Here, low N₂O concentrations across a range of NO₃⁻ concentrations would indicate that these surface sediment areas are relatively rare.

R2-C23: lines 141-142: Stemming from the results of this study, do you have any suggestions of alternative estimation method? What would be need to develop that?

Response:

IPCC uses emission factor (EF_{5-r}) to estimate indirect N₂O emissions from fluvial networks arising from N leaching and runoff, which is simply based on the ratio of dissolved N₂O to NO₃⁻ within streams and rivers. In other words, IPCC only consider a simple linear relationship between NO₃⁻ and N₂O emissions: increases in NO₃⁻ loads to rivers cause increases in N₂O emissions. Apart from NO₃⁻, variation of riverine N₂O concentrations can be also attributed to many other environmental variables, including DO [Rosamond et al., 2012; Venkiteswaran et al., 2014], pH [Audet et al., 2020], stream order [Turner et al., 2015] and so on. These studies examining EF_{5-r} values have suggested calculating the EF_{5-r} using the IPCC protocol may be an oversimplified method. Our regression tree

analysis suggests that “A simple linear NO_3^- model does not adequately predict actual N_2O , because N_2O concentration will not necessarily increase with NO_3^- loads in oxic environments. We therefore recommend that the IPCC methodology should be revised to consider nonlinear relationships or interactions among multiple environmental variables.” (L155-159).

IPCC Climate Change 2007: The Physical Science Basis. (Cambridge Univ. Press, 2007).

Audet, J., et al. (2020), Forest streams are important sources for nitrous oxide emissions, *Global Change Biology*, 26(2), 629-641.

Rosamond, M. S., S. J. Thuss, and S. L. Schiff (2012), Dependence of riverine nitrous oxide emissions on dissolved oxygen levels, *Nature Geoscience*, 5(10), 715-718.

Turner, P. A., T. J. Griffis, X. Lee, J. M. Baker, R. T. Venterea, and J. D. Wood (2015), Indirect nitrous oxide emissions from streams within the US Corn Belt scale with stream order, *Proceedings of the National Academy of Sciences*, 112(32), 9839-9843.

Venkiteswaran, J. J., M. S. Rosamond, and S. L. Schiff (2014), Nonlinear response of riverine N_2O fluxes to oxygen and temperature, *Environmental Science & Technology*, 48(3), 1566-1573.

R2-C24: lines 150-152: Only Yellow River was used here - the warmest catchment with least permafrost and probably highest anthropogenic effect. How typical this is for all the rivers? Please be open about it and discuss how representative this is for the other rivers.

Response:

We do truly regret the poor data availability! The low N_2O yield, and small ratio of *nir/nos* of all sampled rivers provide indications for the widespread occurrence of the complete reduction of N_2O to N_2 . In addition, we reorganized **Supplementary Table 6** to show river reaches in both permafrost-rich and permafrost-poor zones. As displayed in **Supplementary Table 2**, the Upper Yellow Catchment (from MD to TK) is the coldest, and rich in permafrost, which can represent rivers in permafrost-rich zones with very low human disturbance. The Lower Yellow reaches (JG and TNH) are located in permafrost-poor region, representing rivers that are affected by human disturbance.

R2-C25: lines 172-174: Please explain briefly the basic principle of river orders - which order represents the headwaters, which estuary? Also, brief elaboration of “perimeter-to-surface-area ratio and hyporheic exchange rates” would be good to have for permafrost community beyond the

limnologists.

Response:

According to the “top down” system devised by Strahler, streams of the first order are the outermost tributaries. If two streams of the same order merge, the resulting stream is given a number that is one higher. If two rivers with different stream orders merge, the resulting stream is given the higher of the two numbers (see picture below). In line with the stream-river continuum concept, physical variables and biological functionality of a fluvial system change predictably from headwaters to mouth and can be very broadly divided into headwaters (orders 1-3), medium-sized streams (orders 4-6), and large rivers (orders > 6). We have added short elaboration in L188-190 in the main text.

In headwater streams that are typically small and shallow, microbially mediated biogeochemical transformations occurs mainly within the benthic-hyporheic zone. As stream size increases, the ratio of wetted perimeter (red line) to cross-sectional area (light yellow shaded area; see picture below) declines, which reduces the relative contribution of benthic-hyporheic zone to biogeochemical transformations. In large rivers, water column dominates biogeochemical transformations, overwhelming the benthic-hyporheic contribution [Marzadri et al., 2017].

Considering that brief elaboration (L191-193) of “perimeter-to-surface-area ratio and hyporheic

exchange rates” might not be clear enough, we also cite references to assist readers in various fields to help them further understand these terms and principles that are well established in river science.

Marzadri, A., M. M. Dee, D. Tonina, A. Bellin, and J. L. Tank (2017), Role of surface and subsurface processes in scaling N₂O emissions along riverine networks, *Proceedings of the National Academy of Sciences*, 114(17), 4330-4335.

R2-C26: line 258: The correlation between N₂O flux and N₂O concentration, please add a scatter plot.

Response:

As we stated in Supplementary Discussion 1, “A small number of our flux measurements were negative (i.e., N₂O entering the water from the atmosphere), yet all N₂O concentrations were supersaturated. The possible explanation may be that measured N₂O reflect transient concentrations, but N₂O consumption through complete denitrification may occur during 60-min floating chamber deployments.” So, the relationship between N₂O concentrations and fluxes is neither significant nor strong ($P = 0.597$, $R^2 = 0.001$; see below figure), albeit their simultaneous measurements. In contrast, the relationship between N₂O saturation and fluxes is significant but weak ($P = 0.026$, $R^2 = 0.01$; see Supplementary Fig. 5). This suggest gas saturation (O₂ and N₂O) as a normalized metric could be a better indicator in such a mountain environment, because gas solubility changes constantly with altitude.

R2-C27: Fig. 1. The river names indicated with orange text are very poorly readable, please place

them outside the map or add some background for better readability!

Response:

Thanks, we have redrawn this figure.

References:

- Audet, J., et al. (2020), Forest streams are important sources for nitrous oxide emissions, *Global Change Biology*, 26(2), 629-641.
- Begum, M. S., et al. (2021), Localized pollution impacts on greenhouse gas dynamics in three anthropogenically modified Asian river systems, *Journal of Geophysical Research: Biogeosciences*, 126(5), e2020JG006124.
- Harms, T. K., and J. B. Jones Jr. (2012), Thaw depth determines reaction and transport of inorganic nitrogen in valley bottom permafrost soils, *Global Change Biology*, 18(9), 2958-2968.
- Hu, M., D. Chen, and R. A. Dahlgren (2016), Modeling nitrous oxide emission from rivers: A global assessment, *Global Change Biology*, 22(11), 3566-3582.
- Karlsson, J., S. Serikova, S. N. Vorobyev, G. Rocher-Ros, B. Denfeld, and O. S. Pokrovsky (2021), Carbon emission from Western Siberian inland waters, *Nature Communications*, 12(1), 825.
- Khosh, M. S., J. W. McClelland, A. D. Jacobson, T. A. Douglas, A. J. Barker, and G. O. Lehn (2017), Seasonality of dissolved nitrogen from spring melt to fall freezeup in Alaskan Arctic tundra and mountain streams, *Journal of Geophysical Research: Biogeosciences*, 122(7), 1718-1737.
- Kou, D., et al. (2020), Progressive nitrogen limitation across the Tibetan alpine permafrost region, *Nature Communications*, 11(1), 3331.
- Liu, X., et al. (2018), Nitrate is an important nitrogen source for Arctic tundra plants, *Proceedings of the National Academy of Sciences*, 115(13), 3398-3403.
- Marcarelli, A. M., M. A. Baker, and W. A. Wurtsbaugh (2008), Is in-stream N₂ fixation an important N source for benthic communities and stream ecosystems?, *Journal of the North American Benthological Society*, 27(1), 186-211.
- Marzadri, A., M. M. Dee, D. Tonina, A. Bellin, and J. L. Tank (2017), Role of surface and subsurface processes in scaling N₂O emissions along riverine networks, *Proceedings of the National Academy of Sciences*, 114(17), 4330-4335.
- Mishra, U., et al. (2021), Spatial heterogeneity and environmental predictors of permafrost region soil organic carbon stocks, *Science Advances*, 7(9), eaaz5236.
- Mwanake, R. M., G. M. Gettel, K. S. Aho, D. W. Namwaya, F. O. Masese, K. Butterbach-Bahl, and P. A. Raymond (2019), Land use, not stream order, controls N₂O concentration and flux in the Upper Mara River Basin, Kenya, *Journal of Geophysical Research: Biogeosciences*, 124(11), 3491-3506.
- Qu, B., K. S. Aho, C. Li, S. Kang, M. Sillanpää, F. Yan, and P. A. Raymond (2017), Greenhouse gases emissions in rivers of the Tibetan Plateau, *Scientific Reports*, 7(1), 16573.
- Quick, A. M., W. J. Reeder, T. B. Farrell, D. Tonina, K. P. Feris, and S. G. Benner (2019), Nitrous oxide from streams and rivers: A review of primary biogeochemical pathways and environmental variables, *Earth-Science Reviews*, 191, 224-262.
- Rosamond, M. S., S. J. Thuss, and S. L. Schiff (2012), Dependence of riverine nitrous oxide emissions on dissolved oxygen levels, *Nature Geoscience*, 5(10), 715-718.
- Serikova, S., et al. (2018), High riverine CO₂ emissions at the permafrost boundary of Western Siberia, *Nature Geoscience*, 11(11), 825-829.
- Soued, C., P. A. del Giorgio, and R. Maranger (2016), Nitrous oxide sinks and emissions in boreal aquatic networks in Québec, *Nature Geoscience*, 9(2), 116-120.

- Tan, E., W. Zou, Z. Zheng, X. Yan, M. Du, T.-C. Hsu, L. Tian, J. J. Middelburg, T. W. Trull, and S.-j. Kao (2020), Warming stimulates sediment denitrification at the expense of anaerobic ammonium oxidation, *Nature Climate Change*, *10*(4), 349-355.
- Tian, H., et al. (2019), Global soil nitrous oxide emissions since the preindustrial era estimated by an ensemble of terrestrial biosphere models: Magnitude, attribution, and uncertainty, *Global Change Biology*, *25*(2), 640-659.
- Turner, P. A., T. J. Griffis, X. Lee, J. M. Baker, R. T. Venterea, and J. D. Wood (2015), Indirect nitrous oxide emissions from streams within the US Corn Belt scale with stream order, *Proceedings of the National Academy of Sciences*, *112*(32), 9839-9843.
- Venkiteswaran, J. J., M. S. Rosamond, and S. L. Schiff (2014), Nonlinear response of riverine N₂O fluxes to oxygen and temperature, *Environmental Science & Technology*, *48*(3), 1566-1573.
- Voigt, C., M. E. Marushchak, B. W. Abbott, C. Biasi, B. Elberling, S. D. Siciliano, O. Sonnentag, K. J. Stewart, Y. Yang, and P. J. Martikainen (2020), Nitrous oxide emissions from permafrost-affected soils, *Nature Reviews Earth & Environment*, *1*(8), 420-434.
- Wu, W., J. Wang, X. Zhou, B. Yuan, M. Guo, and L. Ren (2020), Spatiotemporal distribution of nitrous oxide (N₂O) emissions from cascade reservoirs in Lancang-Mekong River Yunnan section, Southwestern China, *River Research and Applications*, n/a(n/a).
- Xia, X., L. Zhang, G. Wang, J. Wang, L. Zhang, S. Zhang, and Z. Li (2021), Nitrogen loss from a turbid river network based on N₂ and N₂O fluxes: Importance of suspended sediment, *Science of The Total Environment*, *757*, 143918.
- Yang, X., T. M. Pavelsky, and G. H. Allen (2020), The past and future of global river ice, *Nature*, *577*(7788), 69-73.
- Yao, Y., H. Tian, H. Shi, S. Pan, R. Xu, N. Pan, and J. G. Canadell (2020), Increased global nitrous oxide emissions from streams and rivers in the Anthropocene, *Nature Climate Change*, *10*(2), 138-142.
- Zhang, S., W. Qin, Y. Bai, Z. Zhang, J. Wang, H. Gao, J.-D. Gu, and X. Xia (2021), Linkages between anammox and denitrifying bacterial communities and nitrogen loss rates in high-elevation rivers, *Limnology and Oceanography*, *66*(3), 765-778.

REVIEWERS' COMMENTS

Reviewer #1 (Remarks to the Author):

The authors have diligently addressed all the comments from the reviewers and therefore I recommend this paper for publication.

Minor comments:

L31: "DIN" Abbreviation not introduced before.

Reviewer #2 (Remarks to the Author):

Review report on manuscript by Zhang et al., titled 'Unexpectedly minor nitrous oxide emissions from fluvial networks draining large permafrost catchments of the East Qinghai-Tibet Plateau'

The authors have addressed both reviewers' comments with great care and in their rebuttal letter provide very well argued replies together with some very interesting analysis and clarifications. I read the revised version with pleasure, it is very convincing in the current form. This study has a great potential to act as a game opener, inspiring further studies on N₂O dynamics in water bodies of the rapidly changing permafrost region.

I have only a couple of additional suggestions, that do not require further checks from my side.

Line 93-94: Could it be that this global average is an overestimate due to the lack of data from pristine and northern regions, and plenty of data from regions with intensive human impact? As you discuss on lines 42-44. I see this as one of the main points raising from this study. Is it too speculative to suggest this?

Line 137: Please give that stats also in the text.

Lines 187-188: Could you provide here a reference?

Response to the reviewers' comments

Authors' responses to the reviewers' comments to the manuscript entitled "Unexpectedly minor nitrous oxide emissions from fluvial networks draining large permafrost catchments of the East Qinghai-Tibet Plateau".

We are very grateful to the two reviewers and the editor for approving the revisions that we made during the first revision round and supporting publication of this study! Below we respond to the reviewers' comments with blue font. All line numbers refer to the revised manuscript version with tracked changes. Reviewers' comments are labeled as the reviewer and comment number (e.g., **R1-C1** is the first comment by Reviewer 1).

Response to Reviewer #1

Reviewer #1 (Remarks to the Author):

General comment:

R1-C1: The authors have diligently addressed all the comments from the reviewers and therefore I recommend this paper for publication.

Response:

We would like to thank this reviewer again for the time spent on our study and insightful comments that have greatly improved this paper!

Minor comment:

R1-C2: L31: “DIN” Abbreviation not introduced before.

Response:

Thanks, we have added the definition of “DIN” as “dissolved inorganic nitrogen” the first time we used it. (L30 and 115)

Response to Reviewer #2

Reviewer #2 (Remarks to the Author):

General comment:

R2-C1:

Review report on manuscript by Zhang et al., titled 'Unexpectedly minor nitrous oxide emissions from fluvial networks draining large permafrost catchments of the East Qinghai-Tibet Plateau'.

The authors have addressed both reviewers' comments with great care and in their rebuttal letter provide very well argued replies together with some very interesting analysis and clarifications. I read the revised version with pleasure; it is very convincing in the current form. This study has a great potential to act as a game opener, inspiring further studies on N₂O dynamics in water bodies of the rapidly changing permafrost region.

Response:

First of all, we are happy that this reviewer was satisfied with our revisions to address the earlier comments! Our sincere thanks to the reviewer for the encouraging support and valuable comments that have greatly improved this paper!

Minor comments:

R2-C2:

I have only a couple of additional suggestions, that do not require further checks from my side. Line 93-94: Could it be that this global average is an overestimate due to the lack of data from pristine and northern regions, and plenty of data from regions with intensive human impact? As you discuss on lines 42-44. I see this as one of the main points raising from this study. Is it too speculative to suggest this?

Response:

We do agree with the reviewer's valid point. If we are interested in studying N₂O, we are inclined to sample N-rich streams and rivers with lots of N₂O. In consequence, concentrations below detection limits, and low or even negative fluxes may be missed. Not surprisingly, the geographic distribution of studies is clustered, with heavy representation from agricultural, forested and urban regions and

conspicuous scarcities for vast cryospheric areas of the Siberia, Greenland, Alaska, North Canada, Qinghai-Tibet Plateau, Alps, Antarctic, Andes and so on. This indeed raises the possibility that the current global estimate of average areal N₂O flux may be inflated.

On the other hand, since IPCC's 5th Assessment Report, however, the existing estimates of the magnitude of global fluvial N₂O emissions rose from 32.2 Gg N₂O-N yr⁻¹ in 2016 to ~47.5 Gg N₂O-N yr⁻¹ in 2019, and then to 291.3 Gg N₂O-N yr⁻¹ in 2020 [Hu *et al.*, 2016; Maavara *et al.*, 2019; Yao *et al.*, 2020]. According to this increasing trend, the global average (93.4 μmol m⁻² d⁻¹) quoted from [Hu *et al.*, 2016] is probably not overestimated, but underestimated (assuming that global river and stream surface area remains constant), though there were no reports for global average areal flux in [Maavara *et al.*, 2019; Yao *et al.*, 2020]. The main reason is that small-to-medium sized streams (stream order = 1-4) have been poorly quantified so far, because low-order streams are not consistently gauged for discharge and it is difficult to directly measure their surface area [Turner *et al.*, 2015]. But they contribute outsized amounts of N₂O to the atmosphere (241.4 Gg N₂O-N yr⁻¹) compared to large rivers (stream order ≥ 5; 42.5 Gg N₂O-N yr⁻¹), the latter is comparable to the previous estimates (32.2 Gg N₂O-N yr⁻¹ in 2016 and ~47.5 Gg N₂O-N yr⁻¹ in 2019). In this regard, large N₂O emissions from low-order streams have been ignored or underestimated in earlier estimates of world's river N₂O emissions [Yao *et al.*, 2020]. The conclusion of 'minor N₂O emissions from EQTP rivers' is exactly based on these existing estimates.

Both positive (lack of data from low N systems) and negative (under-sampling small streams) biases in current estimates, we are therefore looking forward to seeing such future studies (including meta-analysis) that tackle this important topic. In addition to N₂O (an intermediate that can be reduced to N₂) measurement, we call for future studies to take N₂ into account, especially for low N systems, so that we can provide a full picture of N₂O emissions. In our study, there exists such a difference, and we hope to emphasize the need to broaden sampling efforts to include less enriched systems and to consider other mechanisms that may shape N₂O dynamics, now and in the future.

Hu, M., et al. (2016), Modeling nitrous oxide emission from rivers: A global assessment, *Global Change Biology*, 22(11), 3566-3582.

Maavara, T., et al. (2019), Nitrous oxide emissions from inland waters: Are IPCC estimates too high? *Global Change Biology*, 25(2), 473-488.

Turner, P. A., et al. (2015), Indirect nitrous oxide emissions from streams within the US Corn Belt scale with stream order, *Proceedings of the National Academy of Sciences*, 112(32), 9839-9843.

Yao, Y., et al. (2020), Increased global nitrous oxide emissions from streams and rivers in the Anthropocene, *Nature Climate Change*, 10(2), 138-142.

R2-C3: Line 137: Please give that stats also in the text.

Response:

We have added the stats to the main text. (L136-137)

R2-C4: Lines 187-188: Could you provide here a reference?

Response:

We have added the relevant reference (i.e., ref. 2) to the sentence (L189):

Quick, A. M., et al (2019), Nitrous oxide from streams and rivers: A review of primary biogeochemical pathways and environmental variables, *Earth-Science Reviews*, 191, 224-262.